# Online Sparsification of Bipartite-Like Clusters in Graphs

Joyentanuj Das [1]   Suranjan De [1]   He Sun [1]

## Abstract

Graph clustering is an important algorithmic technique for analysing massive graphs, and has been widely applied in many research fields of data science. While the objective of most graph clustering algorithms is to find a vertex set of low conductance, a sequence of recent studies highlights the importance of the inter-connection between vertex sets when analysing real-world datasets. Following this line of research, in this work we study bipartite-like clusters and present efficient and online sparsification algorithms that find such clusters in both undirected graphs and directed ones. We conduct experimental studies on both synthetic and real-world datasets, and show that our algorithms significantly speedup the running time of existing clustering algorithms while preserving their effectiveness.

## 1. Introduction

Graph clustering is a fundamental technique in data analysis with wide-ranging applications in machine learning and data science. A classical graph clustering problem involves partitioning the vertices of a graph into sets of highly connected vertices to minimize the normalised cut value. However, many real-world clustering tasks are defined by alternative objective functions, tailored to the specific needs and constraints of the problem at hand. One such example involves uncovering the vertex sets that are densely connected to each other, and these two vertex sets form a bipartite-like graph. For example, when representing the migration or trade datasets with a graph, a bipartite-like cluster captures a typical pattern of regional migration or trade (Cucuringu et al., 2020; Laenen & Sun, 2020; He et al., 2022), and the significance of bipartite-like clusters is further highlighted when studying many other real-world datasets (Bennett et al., 2022; Concas et al., 2022).

[1]School of Informatics, University of Edinburgh, Edinburgh, United Kingdom. Correspondence to: He Sun <h.sun@ed.ac.uk>.

*Proceedings of the 42nd International Conference on Machine Learning*, Vancouver, Canada. PMLR 267, 2025. Copyright 2025 by the author(s).

**Our Results.** We first study bipartite-like clusters in undirected graphs, and present an algorithm that sparsifies an undirected graph while preserving its structure of bipartite-like clusters. Our algorithm can be implemented online, and directly applied to speed up the running time of existing algorithms that find bipartite-like clusters. Formally speaking, for an undirected $G = (V, E)$ and a pair of disjoint and non-empty subsets $A, B \subset V$, we define

$$\overline{\phi}_G(A, B) \triangleq \frac{2w_G(A, B)}{\mathrm{vol}_G(A \cup B)},$$

where

$$w_G(A, B) \triangleq \sum_{\substack{\{u,v\} \in E \\ u \in A, v \in B}} w(u, v)$$

is the cut value between $A$ and $B$ and $\mathrm{vol}_G(A \cup B)$ is the volume of $A \cup B$ defined by

$$\mathrm{vol}_G(A \cup B) = \sum_{\substack{\{u,v\} \in E \\ u \in A \cup B}} w_G(u, v).$$

Notice that a high value of $\overline{\phi}_G(A, B)$ implies that most edges adjacent to the vertices in $A \cup B$ are between $A$ and $B$, and $A$ and $B$ form a bipartite-like cluster. Generalising this to multiple clusters, for every $k \in \mathbb{N}$ we define the $k$-way dual Cheeger constant by

$$\bar{\rho}_G(k) \triangleq \max_{(A_1, B_1), \ldots, (A_k, B_k)} \min_{1 \le i \le k} \overline{\phi}_G(A_i, B_i), \quad (1.1)$$

where the maximum is taken over all the possible $k$ pairs of subsets $(A_1, B_1), \ldots, (A_k, B_k)$ satisfying $A_i \cap A_j = \emptyset, B_i \cap B_j = \emptyset, A_i \cap B_j = \emptyset$ for different $i, j \in [k]$, and $A_i \cup B_i \neq \emptyset$ for different $i, j \in [k]$. Notice that a high value of $\bar{\rho}_G(k)$ implies that $G$ contains $k$ bipartite-like clusters, in each of which the vertex sets $A_i$ and $B_i$ are densely connected to each other. We prove that, when $G$ presents a clear structure of $k$ bipartite-like clusters, this structure can be represented by a sparse subgraph $G^*$ of $G$ with $\widetilde{O}(n)$ edges, and $G^*$ can be constructed online in nearly-linear time[1]. Our result is as follows:

---

[1]We say that a graph algorithm runs in nearly-linear time if the algorithm's running time is $O(m \cdot \mathrm{poly} \log n)$, where $m$ and $n$ are the number of edges and vertices of the input graph. For simplicity, we use $\widetilde{O}(\cdot)$ to hide a poly-logarithmic factor of $n$.

**Theorem 1** (Result for undirected graphs)**.** *Let $G = (V_G, E_G, w_G)$ be an undirected and weighted graph of $m$ edges, and assume that $G$ contains $k$ bipartite-like clusters $(A_1, B_1), \ldots, (A_k, B_k)$ corresponding to $\bar{\rho}_G(k)$. Then, there is an algorithm that runs in $\widetilde{O}(m)$ time and computes a sparsifier $G^* = (V_G, F \subset E_G, \widetilde{w})$, such that these $k$ bipartite-like clusters are preserved in $G^*$ with high probability. That is, it holds with high probability that $\bar{\rho}_{G^*}(k) = \Omega\left(\bar{\rho}_G(k)\right)$, and $G^*$ contains only $k$ bipartite-like clusters.*

Secondly, we study the bipartite-like clusters in directed graphs. Let $\overrightarrow{G} = (V_{\overrightarrow{G}}, E_{\overrightarrow{G}}, w_{\overrightarrow{G}})$ be a digraph with weight function $w_{\overrightarrow{G}} : E_{\overrightarrow{G}} \to \mathbb{R}_{\geq 0}$. For any vertex $u \in V_{\overrightarrow{G}}$, we use $\deg_{\text{out}}(u) \triangleq \sum_{(u,v) \in E} w_G(u, v)$ and $\deg_{\text{in}}(u) \triangleq \sum_{(v,u) \in E} w_G(v, u)$ to express the sum of weights of directed edges with $u$ as the tail or the head, respectively. For any $S \subset V_{\overrightarrow{G}}$, we define $\text{vol}_{\text{out}}(S) \triangleq \sum_{u \in S} \deg_{\text{out}}(u)$ and $\text{vol}_{\text{in}}(S) \triangleq \sum_{u \in S} \deg_{\text{in}}(u)$. For any two disjoint subsets $A, B \subset V_{\overrightarrow{G}}$, we define $\overline{\phi}_{\overrightarrow{G}}(A, B)$ by

$$\overline{\phi}_{\overrightarrow{G}}(A, B) \triangleq \frac{2w_{\overrightarrow{G}}(A, B)}{\text{vol}_{\text{out}}(A) + \text{vol}_{\text{in}}(B)}, \quad (1.2)$$

where

$$w_{\overrightarrow{G}}(A, B) \triangleq \sum_{\substack{(u,v) \in E \\ u \in A, v \in B}} w(u, v)$$

is the sum of the weights of the edges from $A$ to $B$. For every $k \in \mathbb{N}$, the $k$-way directed dual Cheeger constant is defined by

$$\bar{\rho}_{\overrightarrow{G}}(k) \triangleq \max_{(A_1, B_1), \ldots, (A_k, B_k)} \min_{1 \leq i \leq k} \overline{\phi}_{\overrightarrow{G}}(A_i, B_i), \quad (1.3)$$

where the maximum is taken over all the possible $k$ pairs of subsets $(A_1, B_1), \ldots, (A_k, B_k)$ satisfying $A_i \cap A_j = \emptyset, B_i \cap B_j = \emptyset, A_i \cap B_j = \emptyset$ for different $i, j \in [k]$, $A_i \cup B_i \neq \emptyset$ for any $i \in [k]$. By definition, a high value of $\bar{\rho}_{\overrightarrow{G}}(k)$ implies that $\overrightarrow{G}$ contains $k$ bipartite-like clusters $(A_1, B_1), \ldots, (A_k, B_k)$ such that most edges with their tails in $A_i$ have their head in $B_i$ and conversely most edges with their head in $B_i$ have their tail in $A_i$. We prove that, when $\overrightarrow{G}$ presents a structure of $k$ bipartite-like clusters with respect to $\bar{\rho}_{\overrightarrow{G}}(k)$, this structure can be represented by a sparse graph $\overrightarrow{G^*}$ with $\widetilde{O}(n)$ edges, and $\overrightarrow{G^*}$ can be constructed online in nearly-linear time:

**Theorem 2** (Result for directed graphs)**.** *Let $\overrightarrow{G} = (V_{\overrightarrow{G}}, E_{\overrightarrow{G}}, w_{\overrightarrow{G}})$ be a directed and weighted graph of $m$ edges, and assume that $\overrightarrow{G}$ contains $k$ directly bipartite-like clusters $(A_1, B_1), \ldots, (A_k, B_k)$ with respect to $\bar{\rho}_{\overrightarrow{G}}(k)$. Then, there is an algorithm that runs in $\widetilde{O}(m)$ time and computes a sparsifier $\overrightarrow{G^*} = (V_{\overrightarrow{G}}, F \subset E_{\overrightarrow{G}}, \widetilde{w})$, such that*

*these $k$ directed bipartite-like clusters of $\overrightarrow{G}$ are preserved in $\overrightarrow{G^*}$ with high probability. That is, it holds with high probability that $\bar{\rho}_{\overrightarrow{G^*}}(k) = \Omega\left(\bar{\rho}_{\overrightarrow{G}}(k)\right)$, and $\overrightarrow{G^*}$ only contains $k$ directed bipartite-like clusters.*

Now we examine the significance of Theorems 1 and 2. We first highlight that our algorithms preserve the cut values $w(A_i, B_i)$ between the pairs of vertex sets $A_i$ and $B_i$ for $1 \leq i \leq k$; this objective is *different* from the one for most graph sparsification problems, which only preserve the cut values between vertex set $S$ and $V \setminus S$. Secondly, our algorithms preserve $k$ bipartite-like clusters, and the value of $k$ in the output graph is the same as the input graph. Thirdly, our second result works for *directed graphs*; this result is very interesting on its own since most sparsification algorithms are only applicable for undirected graphs. Finally, while the design of most graph sparsification algorithms are based on Laplacian solvers making it unpractical, our designed algorithms only use random sampling.

The design of our algorithms is based on several new reductions and sampling routines, and our algorithms can be implemented online with the degree oracles. As such one can run our algorithms online while exploring the underlying graph with existing local algorithms (e.g., (Andersen, 2010; Li & Peng, 2013)), resulting in direct improvement on the running time of the existing algorithms. To demonstrate this, we conduct experimental studies and show that our algorithms can be directly applied to significantly speed up the running time of the the ones presented in (Macgregor & Sun, 2021a), while preserving similar output results on both the synthetic and real-world datasets.

**Related Work.** Bipartite-like clusters are widely studied in both theoretical computer science and machine learning communities. In theoretical computer science, Trevisan (2009) developed a spectral algorithm that finds a bipartite-like cluster in an undirected graph, and used this to design an approximation algorithm for the max-cut problem. This result is improved by Soto (2015). Liu (2015) studied the relationship between the $k$-way dual Cheeger constant and the eigenvalues of the normalised graph Laplacians, and developed a Cheer-type inequality.

In the machine learning community, bipartite-like clusters are employed to model highly-correlated data items of different types, and algorithms finding these clusters are studied in different settings. Andersen (2010), Li & Peng (2013) and Macgregor & Sun (2021a) presented local algorithms that find bipartite-like clusters, and Macgregor & Sun (2021b) presented an algorithm that finds bipartite components in hypergraphs. Cucuringu et al. (2020) proved that densely connected clusters in a directed graph can be uncovered through spectral clustering on a complex-valued Hermitian matrix representation of directed graphs.

Neumann & Peng (2022) designed a sublinear-time oracle which, under a certain condition, correctly classified the membership of most vertices in a set of hidden planted ground-truth clusters in signed graphs.

Our work relates to finding clusters in *disassortative* networks (Moore et al., 2011; Pei et al., 2019; Zhu et al., 2020), although most existing techniques are based on semi-supervised and global methods. Our work is also related to designing graph sparsification algorithms, e.g., (Spielman & Teng, 2011; Batson et al., 2012; Cohen et al., 2017; Lee & Sun, 2017; 2018). We highlight that, while a spectral sparsifier preserves the cut value $w(S, V \setminus S)$ between any vertex set $S$ and its complement $V \setminus S$, our algorithms' output preserves the cut value $w(A_i, B_i)$ for pairs of vertex sets $A_i$ and $B_i$. Moreover, our algorithms are much easier to implement, and work for *directed* graphs.

## 2. Preliminaries

In this section we list the notation and preliminary results used in the analysis.

**Matrix Representation of Graphs.** We always use $G = (V, E, w)$ to represent an undirected and weighted graph with $n$ vertices and weight function $w : E \to \mathbb{R}_{\geq 0}$. The degree of any vertex $u$ is defined as $d_G(u) = \sum_{u \sim v} w(u, v)$, where the notation $u \sim v$ represents that $u$ and $v$ are adjacent, i.e., $\{u, v\} \in E(G)$. The normalised indicator vector of any $S \subset V$ is defined by

$$\chi_S(v) = \sqrt{\frac{d_G(v)}{\mathrm{vol}_G(S)}}$$

if $v \in S$, and $\chi_S(v) = 0$ otherwise. Let $A_G$ be the adjacency matrix of $G$ defined by $(A_G)_{u,v} = w(u, v)$ if $\{u, v\} \in E(G)$, and $(A_G)_{u,v} = 0$ otherwise. The degree matrix $D_G$ of $G$ is a diagonal matrix defined by $(D_G)_{u,u} = d_G(u)$, and the normalised Laplacian of $G$ is defined by

$$\mathcal{L}_G = I - D_G^{-1/2} A_G D_G^{-1/2}.$$

We can also write the normalised Laplacian matrix with respect to the indicator vectors of the vertices: for each vertex $v$, we define an indicator vector $\chi_v \in \mathbb{R}^n$ by $\chi_v(u) = \frac{1}{\sqrt{d_v}}$ if $u = v$, and $\chi_v(u) = 0$ otherwise. We further define $b_e = \chi_u - \chi_v$ for each edge $e = \{u, v\}$, where the orientation of $e$ is chosen arbitrarily. Then, we have

$$\mathcal{L}_G = \sum_{e=\{u,v\} \in E} w(u, v) \cdot b_e b_e^\mathsf{T}.$$

We also define

$$\mathcal{J}_G \triangleq I + D_G^{-1/2} A_G D_G^{-1/2}.$$

For any symmetric matrix $A \in \mathbb{R}^{n \times n}$, let $\lambda_1(A) \leq \lambda_2(A) \leq \cdots \leq \lambda_n(A)$ be the eigenvalues of $A$. For ease of presentation, we always use $0 = \lambda_1 \leq \lambda_2 \leq \cdots \leq \lambda_n \leq 2$ to express the eigenvalues of $\mathcal{L}_G$, with the corresponding orthonormal eigenvectors $f_1, f_2, \cdots, f_n$. With slight abuse of notation, we use $\mathcal{L}_G^{-1}$ for the pseudo-inverse of $\mathcal{L}_G$, i.e.,

$$\mathcal{L}_G^{-1} \triangleq \sum_{i=2}^n \frac{1}{\lambda_i} f_i f_i^\mathsf{T}.$$

Note that when $G$ is connected, it holds that $\lambda_2 > 0$ and the matrix $\mathcal{L}_G^{-1}$ is well defined. We sometimes drop the subscript $G$ when it is clear from the context.

For any $x \in \mathbb{R}^n$ we define $\|x\| \triangleq \sqrt{\sum_{i=1}^n x_i^2}$, and for any $M \in \mathbb{R}^{n \times n}$ we define

$$\|M\| = \max_{x \in \mathbb{R}^n \setminus \{\mathbf{0}\}} \frac{\|Mx\|}{\|x\|}.$$

**Graph expansion and Cheeger inequality.** For any undirected graph $G$, the expansion (or conductance) of any non-empty subset $S \subset V$ in $G$ is defined as

$$\phi_G(S) \triangleq \frac{w_G(S, \bar{S})}{\mathrm{vol}_G(S)},$$

where $\bar{S}$ is the complement of $S$. We call subsets of vertices $S_1, S_2, \cdots, S_k$ a $k$-way partition of $G$ if $S_i \neq \emptyset$ for all $1 \leq i \leq k$, $S_i \cap S_j = \emptyset$ for $i \neq j$ and $\bigcup_{i=1}^k S_i = V$. For any $k \in \mathbb{N}$, the $k$-way expansion constant is defined as

$$\rho_G(k) = \min_{S_1, S_2, \cdots, S_k} \max_{1 \leq i \leq k} \phi_G(S_i),$$

where the minimum is taken over all possible $k$-way partitions of $G$. Lee et al. (2014) proves the following higher-order Cheeger inequality:

**Lemma 3** (Higher-order Cheeger Inequality, (Lee et al., 2014))**.** *It holds for any undirected graph $G$ of $n$ vertices and integer $1 \leq k \leq n$ that*

$$\lambda_k/2 \leq \rho_G(k) \leq Ck^2 \sqrt{\lambda_k},$$

*where $C$ is a universal constant.*

Generalising this, Liu (2015) proves the following higher-order dual-Cheeger inequality:

**Lemma 4** (Higher-order dual-Cheeger Inequality, (Liu, 2015))**.** *It holds for any undirected graph $G$ of $n$ vertices and integer $1 \leq k \leq n$ that*

$$(2 - \lambda_{n-k+1})/2 \leq 1 - \bar{\rho}_G(k) \leq Ck^3 \sqrt{2 - \lambda_{n-k+1}},$$

*where $C$ is a universal constant.*

The higher-order dual Cheeger inequality can be viewed as a quantitative version of the fact that $\lambda_{n-k+1} = 2$ if and only if $G$ has at least $k$ bipartite connected components.

## 3. Proof of Theorem 1

In this section we present a nearly-linear time sparsification algorithm such that every bipartite-like cluster in an undirected graph $G$ is approximately preserved in the sparsifed graph $G^*$, and sketch the proof. Our result is as follows:

**Theorem 5** (Formal Statement of Theorem 1). *There exists a nearly-linear time algorithm that, given an input graph $G = (V, E, w)$ with $\bar{\rho}_G(k) \geq \frac{1}{\log n}$ for constant some $k$, with high probability computes a sparsifier $G^* = (V, F \subset E, \widetilde{w})$ with $|F| = O\left(\frac{n \cdot \log^3 n}{2 - \lambda_{n-k}}\right)$ edges such that the following hold: (1) $\bar{\rho}_{G^*}(k) = \Omega(\bar{\rho}_G(k))$; (2) $\lambda_{k+1}(\mathcal{J}_{G^*}) = \Theta(\lambda_{k+1}(\mathcal{J}_G))$.*

The first statement of Theorem 5 shows that the $k$ bipartite-like clusters of $G$ is approximately preserved in $G^*$, and together with Lemma 4 the second statement shows that the number of bipartite-like clusters in $G$ and $G^*$ is the same.

**Algorithm.** Our algorithm is similar with (Sun & Zanetti, 2019) at a high level, and is based on sampling edges in $G$ with carefully defined probabilities. Formally, for an input undirected graph $G = (V, E, w_G)$, the algorithm starts with $G^* = (V, \emptyset, \widetilde{w})$ and samples every edge $u \sim v$ in $G$ with probability $p_e \triangleq p_u(v) + p_v(u) - p_u(v) \cdot p_v(u)$, where

$$p_u(v) \triangleq \min\left\{ w_G(u, v) \cdot \frac{C \cdot \log^3 n}{d_G(u) \cdot (2 - \lambda_{n-k})}, 1 \right\}, \quad (3.1)$$

for some constant $C$. For every sampled edge $e = \{u, v\}$, the algorithm adds $e$ to graph $G^*$, and sets $w_{G^*}(e) = w_G(e)/p_e$. Notice that, the choice of $C$ only changes the sampling probability by a constant factor, and doesn't influence the asymptotic order of the sampled edges. Moreover, in practice we usually treat $\frac{C \cdot \log^3 n}{2 - \lambda_{n-k}}$ as $O(\log^c n)$ for a constant $c$, and this only influences the total number of sampled edges and the algorithm's running time by a poly-logarithmic factor.

**Proof Sketch of Theorem 5.** We first prove that the cut values between $A_i$ and $B_i$ in $G$ is preserved in $H$ for any $1 \leq i \leq k$. For any edge $e = \{u, v\}$, we define the random variable $Y_e$ by $Y_e = w_G(u, v)/p_e$ with probability $p_e$, and $Y_e = 0$ otherwise. By defining $X = w_H(A_i, B_i)$, we prove that $\mathbf{E}[X] = w_G(A_i, B_i)$ and

$$\mathbf{E}\left[X^2\right] \leq \frac{2 - \lambda_{n-k}}{C \cdot \log^3 n} \sum_{\substack{e=\{u,v\} \\ u \in A_i, v \in B_i}} w(u, v) \cdot \left(\frac{d_G(u) + d_G(v)}{2}\right).$$

Let $\{(A_i, B_i)\}_{i=1}^k$ be the optimal clusters corresponding to $\bar{\rho}(k)$. Then, we have for every $1 \leq i \leq k$ that

$$\bar{\rho}_G(k) \leq \overline{\phi}_G(A_i, B_i) = \frac{2 w_G(A_i, B_i)}{\mathrm{vol}_G(A_i \cup B_i)},$$

which implies

$$\frac{\bar{\rho}_G(k)}{2} \cdot \mathrm{vol}_G(A_i \cup B_i) \leq \sum_{\substack{e=\{u,v\} \\ u \in A_i, v \in B_i}} w_G(u, v).$$

Applying the Chebyshev's inequality, we have for any constant $c \in \mathbb{R}^+$ that

$$\mathbf{P}\left[|X - \mathbf{E}[X]| \geq c \cdot \mathbf{E}[X]\right] \leq \frac{\mathbf{E}[X^2]}{c^2 \cdot \mathbf{E}[X]^2}$$

$$\leq \frac{2 \cdot (2 - \lambda_{n-k})}{c^2 \cdot C \cdot \log^3 n \cdot \bar{\rho}_G(k)^2}$$

$$\cdot \frac{\left( \max_{\substack{e=\{u,v\} \\ u \in A_i, v \in B_i}} \{d_G(u) + d_G(v)\} \right)}{\mathrm{vol}_G(A_i \cup B_i)^2} \cdot \sum_{\substack{e=\{u,v\} \\ u \in A_i, v \in B_i}} w_G(u, v).$$

Since $\mathrm{vol}_G(A_i \cup B_i) = \sum_{u \in A_i} d_G(u) + \sum_{v \in B_i} d_G(v)$ and $d_G(u) = \sum_{u \sim v} w_G(u, v)$, we have

$$\max_{\substack{e=\{u,v\} \\ u \in A_i, v \in B_i}} (d_G(u) + d_G(v))$$

$$\leq \sum_{u \in A_i} d_G(u) + \sum_{v \in B_i} d_G(v) = \mathrm{vol}_G(A_i \cup B_i)$$

and $\sum_{\substack{e=\{u,v\} \\ u \in A_i, v \in B_i}} w_G(u, v) \leq \mathrm{vol}_G(A_i \cup B_i)$. Applying these gives us that

$$\mathbf{P}\left[|X - \mathbf{E}[X]| \geq c \cdot \mathbf{E}[X]\right]$$

$$\leq \frac{2(2 - \lambda_{n-k})}{c^2 \cdot C \cdot \log^3 n \cdot \bar{\rho}(k)^2} = O\left(\frac{1}{\log n}\right).$$

Hence, by the union bound, we have that $w_H(A_i, B_i) = \Omega(w_G(A_i, B_i))$ for all $1 \leq i \leq k$. The proof of the second statement of Theorem 5 can be found in the appendix. Finally, the total number of edges in $H$ follows by the definition of sampling probability and the Markov inequality. This completes the proof of Theorem 5.

## 4. Proof of Theorem 2

In this section we present a nearly-linear time sparsification algorithm such that every directed bipartite-like cluster in a directed graph is approximately preserved in the output sparsifier, and prove Theorem 2. Specifically, for a digraph $\overrightarrow{G}$ that contains exactly $k$ pairs of $(A_1, B_1), \ldots, (A_k, B_k)$ with high values of $\overline{\phi}_{\overrightarrow{G}}(A_i, B_i)$ for every $1 \leq i \leq k$, our

objective is to construct a sparse digraph $\overrightarrow{G^*}$, such that (i) the values of $\overline{\phi}_{\overrightarrow{G^*}}(A_i, B_i)$ are high for every $1 \leq i \leq k$ and (ii) the number of such pairs in $\overrightarrow{G^*}$ is the same as $\overrightarrow{G}$. Our result is as follows:

**Theorem 6** (Formal Statement of Theorem 2). *There is a nearly-linear time algorithm that, given a directed and weighted graph $\overrightarrow{G} = (V_{\overrightarrow{G}}, E_{\overrightarrow{G}}, w_{\overrightarrow{G}})$ with $n$ vertices and $k$ directed bipartite-like clusters satisfying $\bar{\rho}_{\overrightarrow{G}}(k) = 1 - o(1/k)$ as input, with high probability computes a sparsifier $\overrightarrow{G^*} = (V_{\overrightarrow{G}}, F \subset E_{\overrightarrow{G}}, \widetilde{w})$ such that $\bar{\rho}_{\overrightarrow{G^*}}(k) = \Omega\left(\bar{\rho}_{\overrightarrow{G}}(k)\right)$. Moreover, the total number of edges in the output graph is nearly-linear in $n$.*

Before sketching our technique, recall that, for undirected graphs, the value of $k$ is proven to be identical for $G$ and $G^*$ by analysing the eigenvalues of $\mathcal{J}_G$ and $\mathcal{J}_{G^*}$ and applying the higher-order dual-Cheeger inequality (Lemma 4). However, a natural matrix representation for directed graphs could result in complex-valued eigenvalues, and there is no analogue of Lemma 4 for directed graphs. To overcome this, our developed algorithm is based on a novel reduction from a directed graph to an undirected one, and its reverse operation. Specifically, our designed algorithm consists of the following three steps:

1. for any input digraph $\overrightarrow{G}$, the algorithm constructs an undirected graph $H$ such that every directed bipartite-like cluster defined by $(A_i, B_i)$ in $\overrightarrow{G}$ corresponds to a low-conductance set in $H$;

2. the algorithm constructs a sparsifier $H^*$ of $H$, such that $H$ and $H^*$ have the same structure of clusters;

3. the algorithm applies the sparsified undirected graph $H^*$ to construct a directed graph $\overrightarrow{G^*}$ of $\overrightarrow{G}$ that satisfies $\bar{\rho}_{\overrightarrow{G^*}}(k) = \Omega\left(\bar{\rho}_{\overrightarrow{G}}(k)\right)$.

See Figure 1 for illustration.

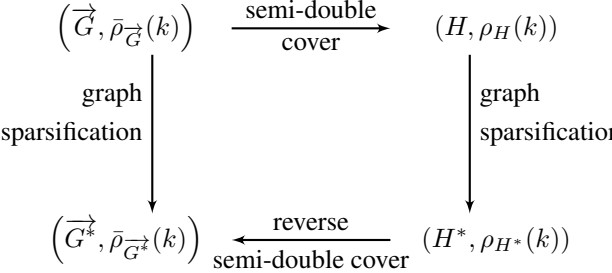

Figure 1: A commutative diagram of our construction. To construct $\overrightarrow{G^*}$ from $\overrightarrow{G}$, we construct graphs $H$ and $H^*$ and prove the close relationships between $\overrightarrow{G}$, $H$, $H^*$, and $\overrightarrow{G^*}$.

**Constructing $H$ from $\overrightarrow{G}$.** Notice that, to preserve $\overline{\phi}_{\overrightarrow{G^*}}(A_i, B_i)$, the cut values $w(A_i, B_i)$ between $A_i$ and $B_i$ need to be approximately preserved in a sparsified directed graph; this objective is different from the most graph sparsification one, which only preserves the cut value between any set $S$ and its complement. To overcome this, we construct an undirected graph $H$ such that every bipartite-like cluster defined by $(A_i, B_i)$ in $\overrightarrow{G}$ corresponds to a low-conductance set in $H$. Specifically, for a weighted digraph $\overrightarrow{G} = (V_{\overrightarrow{G}}, E_{\overrightarrow{G}}, w_{\overrightarrow{G}})$, we construct its semi-double cover $H = (V_H, E_H, w_H)$ as follows: (1) every vertex $v \in V_{\overrightarrow{G}}$ has two corresponding vertices $v_1, v_2 \in V_H$; (2) for every edge $(u, v) \in E_{\overrightarrow{G}}$, we add the edge $\{u_1, v_2\}$ in $E_H$. See Figure 2 for illustration.

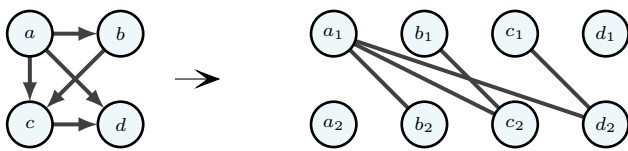

Figure 2: Illustration of the semi-double cover construction. A directed graph of $n$ vertices (left) corresponds to an undirected and bipartite graph of $2n$ vertices (right).

Next we analyse the properties of the reduced graph. Let $\overrightarrow{G}$ be a directed graph with semi-double cover $H$. For any $S \subset V_{\overrightarrow{G}}$, we define $S_1 \subset V_H$ and $S_2 \subset V_H$ by $S_1 \triangleq \{v_1 | v \in S\}$ and $S_2 \triangleq \{v_2 | v \in S\}$. A subset $S$ of $V_H$ is called *simple* if $|\{v_1, v_2\} \cap S| \leq 1$ holds for all $v \in V_{\overrightarrow{G}}$. The following lemma develops a relationship between the *flow ratio* from $A$ to $B$ defined by

$$f_{\overrightarrow{G}}(A, B) \triangleq 1 - \overline{\phi}_{\overrightarrow{G}}(A, B) \tag{4.1}$$

and $\Phi_H(A_1 \cup B_2)$, for any $A, B$.

**Lemma 7.** *Let $\overrightarrow{G}$ be a directed graph with semi-double cover $H$. Then, it holds for any $A, B \subset V_{\overrightarrow{G}}$ that $f_{\overrightarrow{G}}(A, B) = \phi_H(A_1 \cup B_2)$. Similarly, for any simple set $S \subset V_H$, let $A = \{u : u_1 \in S\}$ and $B = \{u : u_2 \in S\}$. Then, it holds that $f_{\overrightarrow{G}}(A, B) = \phi_H(S)$.*

Lemma 7 proves a one-to-one correspondence between any bipartite-like cluster in $\overrightarrow{G}$ and a vertex set in $H$. Building on this, we prove that this one-to-one correspondence can be generalised between any $k$ bipartite-like clusters in $\overrightarrow{G}$ and $k$ disjoint vertex sets in $H$. Moreover, the structure of $k$ bipartite-like clusters in $\overrightarrow{G}$ is preserved by a collection of $k$ disjoint vertex sets of low conductance in $H$.

**Lemma 8.** *For any directed and weighted graph $\overrightarrow{G} = (V_{\overrightarrow{G}}, E_{\overrightarrow{G}}, w_{\overrightarrow{G}})$ and $k \in \mathbb{N}$, it holds that*

$$\bar{\rho}_{\overrightarrow{G}}(k) = 1 - \min_{C_1, \ldots, C_k} \max_{1 \leq i \leq k} \phi_H(C_i), \tag{4.2}$$

*where the minimum is taken over $k$ disjoint simple subsets of $V_H$ defined by $C_i = A_{i_1} \cup B_{i_2}$ for $1 \le i \le k$.*

**Sparsification of $H$.** Next we construct a sparse representation of $H$, denoted by $H^*$, such that the $k$ vertex sets of low conductance is preserved in $H^*$. To achieve this, we apply the following result to construct a cluster-preserving sparsifier.

**Lemma 9** ((Sun & Zanetti, 2019)). *There exists a nearly-linear time algorithm that, given a graph $G = (V, E, w)$ with $k$ clusters as input, with probability at least $9/10$, computes a sparsifier $H = (V, F \subset E, \widetilde{w})$ with $|F| = O((1/\lambda_{k+1}) \cdot n \log n)$ edges such that the following holds: (1) it holds for any $1 \le i \le k$ that $\phi_H(S_i) = O(k \cdot \phi_G(S_i))$, where $S_1, \cdots, S_k$ are the optimal clusters in $G$ that achieves $\rho(k)$; (2) $\lambda_{k+1}(\mathcal{L}_H) = \Omega(\lambda_{k+1}(\mathcal{L}_G))$.*

**Constructing $\overrightarrow{G^*}$ from $H^*$.** Finally, we construct a directed graph $\overrightarrow{G^*}$ from $H^*$ such that the original $k$ directed bipartite-like clusters in $\overrightarrow{G}$ is preserved in $\overrightarrow{G^*}$. To achieve this, we introduce the following *reverse semi-double cover*:

**Definition 10** (reverse semi-double cover). *Given any double cover graph $H^* = (V_{H^*}, E_{H^*}, w_{H^*})$ as input, the reverse semi-double cover of $H^*$ is a directed graph $\overrightarrow{G^*} = (V_{\overrightarrow{G^*}}, E_{\overrightarrow{G^*}}, w_{\overrightarrow{G^*}})$ constructed as follows:*

- *every pair of vertices $u_1$ and $u_2$ in $V_{H^*}$ corresponds to a vertex $v \in V_{\overrightarrow{G^*}}$;*

- *we add an edge $(u, v)$ to $E_{\overrightarrow{G}}$ if there is edge $\{u_1, v_2\} \in E_{H^*}$, and set $w_{\overrightarrow{G^*}}(u, v) = w_{H^*}(u_1, v_2)$.*

One might think that the reverse double cover plays an exact opposite role of the double cover, however it is not the case. In particular, while our constructed subsets $C_1, \ldots, C_k$ in the first step are always simple in $H$ (cf. Lemma 8), the $k$ subsets corresponding to $\rho_H(k)$ are not necessarily simple. As a result,

$$\min_{C_1, \ldots, C_k} \max_{1 \le i \le k} \phi_H(C_i) = \rho_H(k)$$

doesn't hold in general, and there is no direct correspondence between $C_1, \ldots, C_k$ in $H$ and the $k$ directed bipartite-like clusters in $\overrightarrow{G^*}$ that correspond to $\bar{\rho}_{\overrightarrow{G^*}}(k)$.

To analyse $\rho_{\overrightarrow{G^*}}(k)$, for any set $S \subset V_H$ we partition the set into two subsets $S_1$ and $S_2$ defined by $S_1 = S \cap (A_{i_1} \cup B_{i_2})$ and $S_2 = S \cap (A_{i_2} \cup B_{i_1})$. For example, following Figure 2, if $A_i = \{a, c\}$ and $B_i = \{b, d\}$ and the set $S \subset V_H$ is $S = \{a_1, b_1, b_2, c_1, c_2\}$, then we have $S_1 = \{a_1, b_2, c_1\}$ and $S_2 = \{b_1, c_2\}$. As $A_i$ and $B_i$ are densely connected in $H$, there are few edges within $A_i$ and $B_i$ for $1 \le i \le k$.

Hence, there are very few edges between $S_1$ and $S_2$ for any $S \subset V_H$. Without loss of generality, we assume that

$$\frac{2w_H(S_1, S_2)}{w_H(S_1, \bar{S}_1) + w_H(S_2, \bar{S}_2)} \le c$$

for some constant $c < 1$. Simplifying the inequality above we get

$$w_H(S_1, \bar{S}_1) + w_H(S_2, \bar{S}_2) - 2w_H(S_1, S_2)$$
$$\ge (1 - c) \cdot \left[ w_H(S_1, \bar{S}_1) + w_H(S_2, \bar{S}_2) \right].$$

Thus, for any not necessarily simple vertex set $S \subset V_H$ we have

$$\phi_H(S) = \frac{w_H(S, \bar{S})}{\mathrm{vol}(S)}$$
$$= \frac{w_H(S_1, \bar{S}_1) + w_H(S_2, \bar{S}) - 2w_H(S_1, S_2)}{\mathrm{vol}(S_1) + \mathrm{vol}(S_2)}$$
$$\ge (1 - c) \cdot \min \left\{ \frac{w_H(S_1, \bar{S}_1)}{\mathrm{vol}(S_1)}, \frac{w_H(S_2, \bar{S}_2)}{\mathrm{vol}(S_2)} \right\}$$
$$= (1 - c) \cdot \min \left\{ \phi_H(S_1), \phi_H(S_2) \right\},$$

where the last inequality follows by the median inequality. Thus, for every $S \subset V_H$, there is a simple set $T \subset V_H$ such that $\phi_H(S) \ge (1 - c) \cdot \phi_H(T)$. Moreover, for any collection of $k$-disjoint sets $S_1, S_2, \cdots, S_k$, where $S_i \subset V_H$ we have a collection of $k$-disjoint simple sets $T_1, T_2, \cdots, T_k$, where $T_i \subset V_H$, such that

$$\max_{1 \le i \le k} \phi_H(S_i) \ge (1 - c) \cdot \max_{1 \le i \le k} \phi_H(T_i).$$

Taking minimum over all such collection of $k$-disjoint subsets of $V_H$ gives us that

$$\min_{S_1, S_2, \cdots, S_k} \max_{1 \le i \le k} \phi_H(S_i)$$
$$= \rho_H(k) \ge (1 - c) \cdot \min_{T_1, T_2, \cdots, T_k} \max_{1 \le i \le k} \phi_H(T_i),$$

where in the second half of the inequality the minimum is taken over collection of $k$-disjoint simple subsets of $V_H$. On one hand, rearranging the above inequality we have

$$\frac{1}{1 - c} \cdot \rho_H(k) \ge \min_{T_1, T_2, \cdots, T_k} \max_{1 \le i \le k} \phi_H(T_i), \qquad (4.3)$$

and on the other hand, since the collection of $k$-disjoint simple subsets of $V_H$ is a sub-collection of the collection of $k$-disjoint subsets of $V_H$, we have

$$\min_{T_1, T_2, \cdots, T_k} \max_{1 \le i \le k} \phi_H(T_i) \ge \rho_H(k). \qquad (4.4)$$

Thus, combining (4.3) and (4.4), we have

$$\frac{1}{1 - c} \cdot \rho_H(k) \ge \min_{T_1, T_2, \cdots, T_k} \max_{1 \le i \le k} \phi_H(T_i) \ge \rho_H(k). \qquad (4.5)$$

Further, combining (4.2) and (4.5) we have

$$1 - \frac{1}{1 - c} \cdot \rho_H(k) \le \bar{\rho}_{\overrightarrow{G}}(k) \le 1 - \rho_H(k). \qquad (4.6)$$

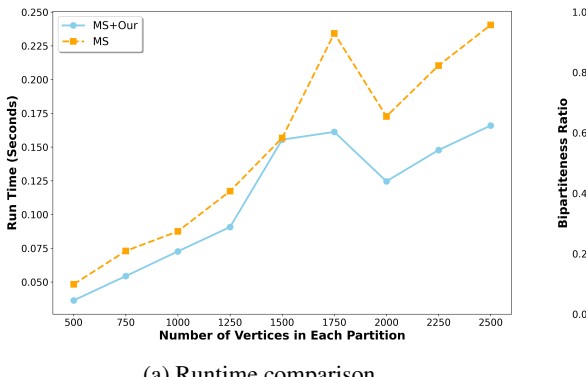

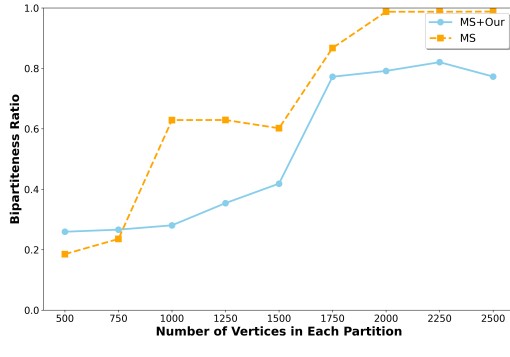

(a) Runtime comparison

(b) Bipartiteness Ratio comparison

Figure 3: Runtime and bipartiteness comparison between MS and our algorithm by fixing $p = 0.3$, $q = 0.1p$ and varying the number of vertices between $500$ and $2,500$ in each partition.

**Proof of Theorem 2.** Now we are ready to prove Theorem 2. Since $\overrightarrow{G}$ is a directed graph with $k$ bipartite-like clusters, the value of $\bar{\rho}_{\overrightarrow{G}}(k)$ is high; together with (4.6), this implies that $\rho_H(k) = o(1)$. By Lemma 9, we know that there exists a sparsifier $H^*$ of $H$, such that $\rho_{H^*}(k) = O(k \cdot \rho_H(k))$. Thus, we can conclude that $\rho_{H^*}(k) = o(1)$. Hence, applying (4.6) for $\overrightarrow{G^*}$ and $H^*$ we have

$$1 - \frac{1}{1-c} \cdot \rho_{H^*}(k) \leq \bar{\rho}_{\overrightarrow{G^*}}(k) \leq 1 - \rho_{H^*}(k). \quad (4.7)$$

Finally, using the fact that $\rho_{H^*}(k) = o(1)$, we conclude that $\bar{\rho}_{\overrightarrow{G^*}}(k)$ is close to $1$ and hence the structure of $\overrightarrow{G}$ will be preserved in $\overrightarrow{G^*}$. Moreover, by the construction of $H$, and $H^*$, and $\overrightarrow{G^*}$, the value of $k$ is preserved.

For the running time, notice that all the intermediate graphs $H$ and $H^*$ can be constructed locally, and it's sufficient to examine every edge of the input graph $\overrightarrow{G}$ once throughout the execution of the algorithm. This implies the nearly-linear running time of our overall algorithm. Combining everything above above proves Theorem 2.

# 5. Experiments

We evaluate the performance of our proposed algorithms on synthetic and real-world datasets. We employ the algorithms presented in (Macgregor & Sun, 2021a) as the baseline algorithms, and examine the speedup of their algorithms when applying our sparsification algorithms as subroutines. Notice that, as all the involved operations of our algorithms can be performed locally, one can run our graph sparsification algorithms online while exploring the underlying graph with a local algorithm. For ease of presentation, in this section we call the local algorithm in (Macgregor & Sun, 2021a) with our sparsification framework our algorithm.

All experiments were performed on a HP ZBook Studio with 11th Gen Intel(R) Core(TM) i7-11800H @ 2.30GHz processor and 32 GB of RAM. Our code can be downloaded from https://github.com/suranjande4/Online-Sparsification-of-Bipartite-Like-Clusters-in-Graphs.

## 5.1. Results for Undirected Graphs

**Synthetic Dataset.** We compare the performance of our algorithm with the LOCBIPARTDC algorithm presented in (Macgregor & Sun, 2021a), which we refer to as MS, on synthetic graphs generated from the stochastic block model (SBM). Specifically, we assume that the graph has $k = 2$ clusters, say $C_1, C_2$, and the number of vertices in each cluster, denoted by $n_1$ and $n_2$ respectively, satisfies $n_1 = n_2$. Moreover, any pair of vertices $u \in C_i$ and $v \in C_j$ is connected with probability $p_{ij}$. We assume that $p_{12} = p_{21} = p$ and $p_{11} = p_{22} = q$, where $q = 0.1p$. Throughout the experiments, we leave the parameters $n$ and $p$ free but maintain the above relations.

Our algorithm sparsifies the underlying graph and simultaneously applies the MS algorithm. We evaluate the quality of the output $(L, R)$ returned by each algorithm with respect to its bipartiteness ratio defined by $\beta(L, R) = 1 - \bar{\phi}(L, R)$. All our reported results are the average performance of each algorithm over 10 runs, in which a random vertex from $C_1 \cup C_2$ is chosen as the starting vertex of the algorithm. We generate graphs from the SBM such that $q = 0.1p$ and vary the size of the target set by varying $n_1$ between $500$ and $2,500$. In Figure 3, we fix the probability $p = 0.3$ and vary the number of vertices $n_1 = n_2$ and compare both runtime and the bipartiteness ratio between the MS algorithm and our algorithm. One can observe that for a fixed probability $p$ as we increase the number of vertices, our algorithm takes much less time than the MS algorithm and maintains a similar bipartiteness ratio with the MS algorithm.

Table 1: Comparison of MS with our algorithm on the Militarised Interstate Disputes Dataset. We use the vertices corresponding to the listed countries in the first column as the seed vertex of the local algorithm.

| COUNTRY NAME | MS RUNTIME | MS BIPARTITENESS | OUR ALGO. RUNTIME | OUR ALGO. BIPARTITENESS |
|---|---|---|---|---|
| USA | 0.034 | 0.292 | 0.0044 | 0.285 |
| NETHERLANDS | 0.0351 | 0.307 | 0.0042 | 0.281 |
| LITHUANIA | 0.0336 | 0.303 | 0.0043 | 0.165 |

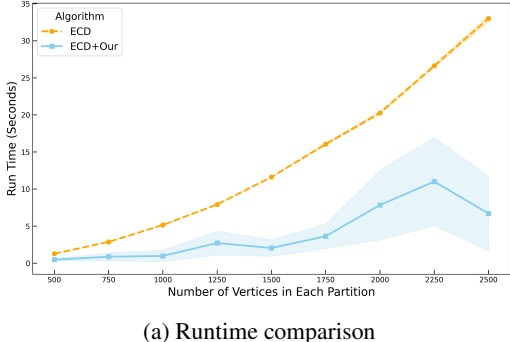

(a) Runtime comparison

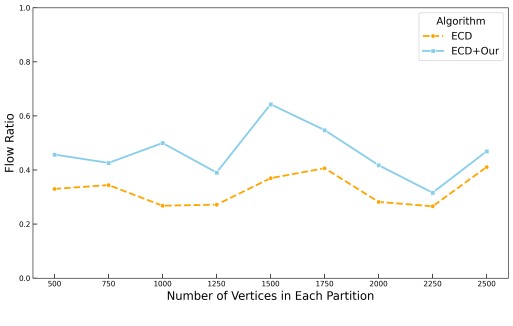

(b) Flow-ratio comparison

Figure 4: Runtime and flow-ratio comparison between ECD and our algorithm by fixing $\eta = 0.7$ and varying the number of vertices in each partition from $500$ to $2,500$.

**Real-world Dataset.** We evaluate the performance of our algorithm on the Dyadic Militarised Interstate Disputes Dataset (v3.1) (Maoz et al., 2019), which records every interstate dispute during 1900–1950, including the level of hostility resulting from the dispute and the number of casualties. We construct a graph from the dataset as follows: every country is represented by a vertex; two vertices are connected by an edge with weight 30 if there is a war between the corresponding countries, and the two vertices are connected by an edge with weight 1 if the corresponding countries have other dispute which is not part of an interstate war. We set $\gamma = 0.02$ for the MS algorithm, and Table 1 compares the performance of the MS algorithm with ours. This shows that our algorithm takes much less time than the MS algorithm and maintains a similar bipartiteness ratio.

### 5.2. Results for Directed Graphs

**Synthetic Dataset.** We compare the performance of our algorithm with the EVOCUTDIRECTED algorithm presented in (Macgregor & Sun, 2021a), which we refer to as ECD, and use the graphs generated from the SBM as the algorithms' input. In our algorithm, given a digraph $G$ as input, we sparsify the graph along with generating the volume-biased ESP on $G's$ semi-double cover $H$. Since the ECD is a local algorithm, we also test our algorithm locally. In this model, we look into a cluster which is almost bipartite with the bipartition being $L$ and $R$. We set the number of vertices in $L$ and $R$ to be $n_1$ and $n_2$ such that $n_1 = n_2$ and

the probability of assigning an edge is defined by

$$\begin{array}{cc} & \begin{array}{cc} L & R \end{array} \\ \begin{array}{c} L \\ R \end{array} & \begin{pmatrix} 9/n_1 & \eta \\ 1-\eta & 9/n_2 \end{pmatrix}, \end{array}$$

i.e., the probability that there is an edge within the partition is $9/n_1 = 9/n_2$ and so on. As most of our directed edges are from $L$ to $R$, the value of $\eta$ is high. For our experiments we generate two sets of plots:

- We fix the value of $\eta = 0.7$ and increase the number of vertices in each partition from $500$ to $2,500$, and compare the runtime of ECD and our algorithm. As reported in Figure 4, our algorithm takes much less time than the ECD algorithm and gives a similar flow-ratio at the same time as we increase the number of vertices.

- We increase the number of vertices in each partition from $500$ to $2,500$, increase the value of $\eta$ from $0.7$ to $0.9$, and compare the runtime of the ECD algorithm and our algorithm. As reported in Figure 5, our algorithm runs faster than the ECD algorithm as $\eta$ increases.

**Real-world Dataset.** Now we evaluate the algorithms' performance on the US Migration Dataset (U.S. Census Bureau, 2000). We construct the digraph from the dataset as follows: every county in the mainland USA is represented

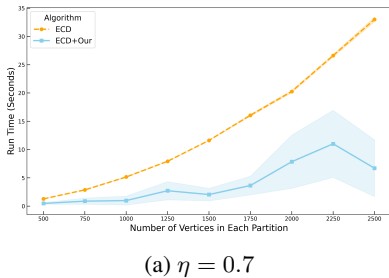

(a) $\eta = 0.7$

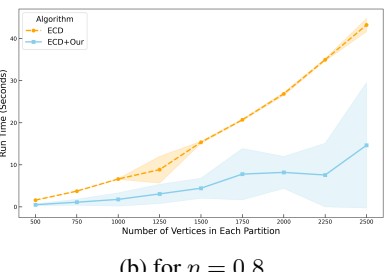

(b) for $\eta = 0.8$

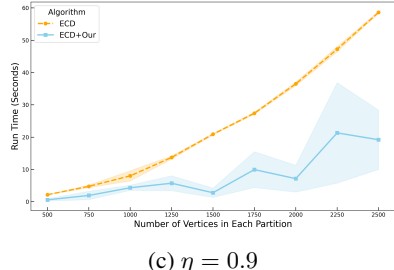

(c) $\eta = 0.9$

Figure 5: Runtime comparison between ECD and our algorithm for $\eta = 0.7, 0.8$, and $0.9$ respectively.

Table 2: Comparison of ECD with our algorithm on real-world datasets. We use the vertices corresponding to the listed counties in the first column as the seed vertex of the local algorithm.

| COUNTY NAME | TARGET $\phi$ | ECD RUNTIME | ECD FLOW-RATIO | OUR ALGO. RUNTIME | OUR ALGO. FLOW-RATIO |
|---|---|---|---|---|---|
| Maricopa County | 0.2 | 20.661 | 0.414 | 13.434 | 0.417 |
| Virginia Beach City | 0.2 | 15.31 | 0.546 | 12.29 | 0.621 |
| Kanawha county | 0.2 | 9.318 | 0.33 | 8.483 | 0.33 |

by a vertex; for any vertices $i, j$, the edge weight of is given by $|(M_{i,j} - M_{j,i})/(M_{i,j} + M_{j,i})|$, where $M_{i,j}$ is the number of people who migrated from county to county between 1995 and 2000; in addition, the direction of $(i, j)$ is set to be from $i$ to $j$ if $M_{i,j} > M_{j,i}$, otherwise the direction is set to be the opposite. We compare the output of ECD and the output of ECD when applying our sparsification algorithm as subroutine. Furthermore, we use the vertices corresponding to different counties as the input of the local algorithm ECD. As shown in Table 2, with our developed algorithm the local ECD algorithm achieves roughly the same flow ratio, and our sparsification procedure significantly speeds up the total running time of the algorithm. Moreover, the runtime speedup is more significant when the local algorithm explores more parts of the input graph.

In conclusion, these experimental studies demonstrate that our developed algorithms can be directly employed to speed up the running time of existing algorithms that find bipartite-like clusters, and can be widely applied when analysing datasets of various domains. We believe that our developed techniques and algorithms will motivate future and fruitful studies for analysing complex cluster structures of graphs.

## Impact Statement

This paper presents work whose goal is to advance the field of Machine Learning. There are many potential societal consequences of our work, none of which we feel must be specifically highlighted here.

## Acknowledgements

The first and third authors of the paper are supported by EPSRC Early Career Fellowship (EP/T00729X/1).

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

## A. Useful Inequalities

The following inequalities will be used in our analysis.

**Theorem 11** (Courant-Fischer Theorem). *Let $A$ be a $n \times n$ symmetric matrix with eigenvalues $\lambda_1 \leq \lambda_2 \leq \cdots \leq \lambda_n$. Then, it holds for any $1 \leq k \leq n$ that*

$$\lambda_k = \min_{\substack{S \\ \dim(S)=k}} \max_{y \in S \setminus \{\mathbf{0}\}} \frac{y^\mathsf{T} \cdot A \cdot y}{y^\mathsf{T} \cdot y}$$

$$= \max_{\substack{S \\ \dim(S)=n-k+1}} \min_{y \in S \setminus \{\mathbf{0}\}} \frac{y^\mathsf{T} \cdot A \cdot y}{y^\mathsf{T} \cdot y},$$

*where the maximisation and minimisation are over the subspaces of $\mathbb{R}^n$.*

**Lemma 12** (Bernstein's Inequality). *Let $X_1, X_2, \cdots, X_n$ be independent random variables such that $|X_i| \leq M$ for any $1 \leq i \leq n$. Let $X = \sum_{i=1}^n X_i$, and $R = \sum_{i=1}^n \mathbf{E}\left[X_i^2\right]$. Then, it holds that*

$$\mathbf{P}\left[|X - \mathbf{E}[X]| \geq t\right] \leq 2\exp\left(-\frac{t^2}{2\left(R + \frac{Mt}{3}\right)}\right).$$

**Lemma 13** (Matrix Chernoff Bound). *Consider a finite sequence $\{X_i\}$ of independent, random, PSD matrices of dimension $d$ that satisfy $\|X_i\| \leq R$. Let $\mu_{\min} = \lambda_{\min}\left(\mathbf{E}[\sum_i X_i]\right)$ and $\mu_{\max} = \lambda_{\max}\left(\mathbf{E}[\sum_i X_i]\right)$. Then, it holds that*

$$\mathbf{P}\left[\lambda_{\min}\left(\sum_i X_i\right) \leq (1-\delta)\mu_{\min}\right] \leq d \cdot \left(\frac{e^{-\delta}}{(1-\delta)^{1-\delta}}\right)^{\frac{\mu_{\min}}{R}},$$

*for $\delta \in [0,1]$, and*

$$\mathbf{P}\left[\lambda_{\max}\left(\sum_i X_i\right) \geq (1+\delta)\mu_{\max}\right] \leq d \cdot \left(\frac{e^{\delta}}{(1+\delta)^{1+\delta}}\right)^{\frac{\mu_{\max}}{R}}$$

*for $\delta \geq 0$.*

## B. Omitted Detail from Section 3

This section presents all the omitted detail from Section 3, and gives a complete proof of Theorem 5. We first recall that, for every vertex $u$ and its adjacent vertex $v$, the algorithm assigns the edge $e = \{u, v\}$ the probability

$$p_u(v) \triangleq \min\left\{w_G(u,v) \cdot \frac{C \cdot \log^3 n}{d_G(u) \cdot (2 - \lambda_{n-k})}, 1\right\}, \tag{B.1}$$

for a large enough constant $C \in \mathbb{R}_{\geq 0}$. The algorithm checks every edge and samples an edge $e = \{u, v\}$ with probability $p_e$, where

$$p_e \triangleq p_u(v) + p_v(u) - p_u(v) \cdot p_v(u).$$

Note that, it is easy to check that $p_e$ satisfies the inequality

$$\frac{1}{2}(p_u(v) + p_v(u)) \leq p_e \leq p_u(v) + p_v(u).$$

We start with an empty set $F$ and gradually store all the sampled edges in $F$, which is sampled by the algorithm. Finally, the algorithm returns a weighted graph $H = (V, F, w_H)$, where the weight $w_H(u,v)$ of every sampled edge $e = \{u, v\} \in F$ is defined by

$$w_H(u,v) = \frac{w_G(u,v)}{p_e}.$$

Next, we analyze the size of $F$. Since

$$\sum_u \sum_{e=\{u,v\}} w_G(u,v) \cdot \frac{C \cdot \log^3 n}{d_G(u) \cdot (2 - \lambda_{n-k})} = O\left(\frac{n \cdot \log^3 n}{2 - \lambda_{n-k}}\right),$$

it holds by Markov inequality that with constant probability the number of edges $e = \{u, v\}$ with $p_u(v) \geq 1$ is $O\left(\frac{n \cdot \log^3 n}{2 - \lambda_{n-k}}\right)$. Without loss of generality, we assume that these edges are in $F$, and in the remaining part of the proof we assume it holds for any edge $u \sim v$ that

$$w_G(u, v) \cdot \frac{C \cdot \log^3 n}{d_G(u) \cdot (2 - \lambda_{n-k})} < 1.$$

Then, the expected number of edges in $H$ equals

$$\sum_{e=\{u,v\}} p_e \leq \sum_{e=\{u,v\}} p_u(v) + p_v(u)$$

$$= \frac{C \cdot \log^3 n}{(2 - \lambda_{n-k})} \sum_{e=\{u,v\}} w(u, v) \cdot \left(\frac{1}{d_G(u)} + \frac{1}{d_G(v)}\right)$$

$$= O\left(\frac{n \cdot \log^3 n}{2 - \lambda_{n-k}}\right),$$

and by Markov inequality it holds with constant probability that

$$|F| = O\left(\frac{n \cdot \log^3 n}{2 - \lambda_{n-k}}\right).$$

Now we show that the cut value between $A_i$ and $B_i$ is preserved in $H$ for all $1 \leq i \leq k$. For any edge $e = \{u, v\}$, we define the random variable $Y_e$ by

$$Y_e = \begin{cases} \dfrac{w_G(u, v)}{p_e} & \text{with probability } p_e, \\ 0 & \text{otherwise.} \end{cases} \tag{B.2}$$

Also, we define $X = w_H(A_i, B_i)$, and have that

$$\mathbf{E}[X] = \sum_{\substack{e=\{u,v\} \\ u \in A_i, v \in B_i}} \mathbf{E}[Y_e] = \sum_{\substack{e=\{u,v\} \\ u \in A_i, v \in B_i}} p_e \cdot \frac{w_G(u, v)}{p_e}$$

$$= \sum_{\substack{e=\{u,v\} \\ u \in A_i, v \in B_i}} w_G(u, v) = w_G(A_i, B_i). \tag{B.3}$$

Next, we analyse the second moment of the random variable $X$ and have that

$$\mathbf{E}\left[X^2\right] = \sum_{\substack{e=\{u,v\} \\ u \in A_i, v \in B_i}} p_e \cdot \left(\frac{w_G(u, v)}{p_e}\right)^2$$

$$= \sum_{\substack{e=\{u,v\} \\ u \in A_i, v \in B_i}} \frac{w_G(u, v)^2}{p_e}$$

$$\leq \sum_{\substack{e=\{u,v\} \\ u \in A_i, v \in B_i}} \frac{2 w_G(u, v)^2}{p_u(v) + p_v(u)} \tag{B.4}$$

$$= \sum_{\substack{e=\{u,v\} \\ u \in A_i, v \in B_i}} \frac{2 w_G(u, v)^2}{\frac{w_G(u,v) \cdot C \cdot \log^3 n}{(2 - \lambda_{n-k})} \cdot \left(\frac{1}{d_G(u)} + \frac{1}{d_G(v)}\right)}$$

$$\leq \frac{2 - \lambda_{n-k}}{C \cdot \log^3 n} \sum_{\substack{e=\{u,v\} \\ u \in A_i, v \in B_i}} w(u, v) \cdot \left(\frac{d_G(u) + d_G(v)}{2}\right),$$

where the last step follows by the means inequality. Let $\{(A_i, B_i)\}_{i=1}^k$ be the optimal cluster where $\bar{\rho}(k)$ is attained for graph $G$. Recall that for every $k \in \mathbb{N}$, the $k$-way dual Cheeger constant is defined by

$$\bar{\rho}_G(k) = \max_{(A_1,B_1),\cdots,(A_k,B_k)} \min_{1 \le i \le k} \overline{\phi}_G(A_i, B_i).$$

Then, we have for every $1 \le i \le k$ that

$$\bar{\rho}_G(k) \le \overline{\phi}_G(A_i, B_i) = \frac{2w_G(A_i, B_i)}{\text{vol}_G(A_i \cup B_i)},$$

which implies

$$\frac{\bar{\rho}_G(k)}{2} \cdot \text{vol}_G(A_i \cup B_i) \le \sum_{\substack{e=\{u,v\} \\ u \in A_i, v \in B_i}} w_G(u, v). \tag{B.5}$$

Next, by the Chebyshev's inequality we have for any constant $c \in \mathbb{R}^+$ that

$$\mathbf{P}\left[|X - \mathbf{E}[X]| \ge c \cdot \mathbf{E}[X]\right]$$

$$\le \frac{\mathbf{E}[X^2]}{c^2 \cdot \mathbf{E}[X]^2}$$

$$\le \frac{\frac{2-\lambda_{n-k}}{C \cdot \log^3 n}\left(\sum_{\substack{e=\{u,v\} \\ u \in A_i, v \in B_i}} w_G(u, v) \cdot \left(\frac{d_G(u)+d_G(v)}{2}\right)\right)}{c^2 \cdot \left(\sum_{\substack{e=\{u,v\} \\ u \in A_i, v \in B_i}} w_G(u, v)\right)^2}$$

$$\le \frac{\frac{2-\lambda_{n-k}}{C \cdot \log^3 n}\left(\sum_{\substack{e=\{u,v\} \\ u \in A_i, v \in B_i}} w_G(u, v) \cdot \left(\frac{d_G(u)+d_G(v)}{2}\right)\right)}{c^2 \cdot \left(\frac{\bar{\rho}_G(k)}{2} \cdot \text{vol}_G(A_i \cup B_i)\right)^2} \tag{B.6}$$

$$= \frac{2 \cdot (2 - \lambda_{n-k})}{c^2 \cdot C \cdot \log^3 n \cdot \bar{\rho}_G(k)^2} \cdot \frac{\sum_{\substack{e=\{u,v\} \\ u \in A_i, v \in B_i}} w_G(u, v) \cdot (d_G(u) + d_G(v))}{\text{vol}_G(A_i \cup B_i)^2}$$

$$\le \frac{2 \cdot (2 - \lambda_{n-k})}{c^2 \cdot C \cdot \log^3 n \cdot \bar{\rho}_G(k)^2} \cdot \left(\max_{\substack{e=\{u,v\} \\ u \in A_i, v \in B_i}} \{d_G(u) + d_G(v)\}\right) \cdot \frac{\sum_{\substack{e=\{u,v\} \\ u \in A_i, v \in B_i}} w_G(u, v)}{\text{vol}_G(A_i \cup B_i)^2}.$$

Since $\text{vol}_G(A_i \cup B_i) = \sum_{u \in A_i} d_G(u) + \sum_{v \in B_i} d_G(v)$ and $d_G(u) = \sum_{u \sim v} w_G(u, v)$, we have

$$\max_{\substack{e=\{u,v\} \\ u \in A_i, v \in B_i}} \{d_G(u) + d_G(v)\} \le \sum_{u \in A_i} d_G(u) + \sum_{v \in B_i} d_G(v)$$

$$= \text{vol}_G(A_i \cup B_i)$$

and

$$\sum_{\substack{e=\{u,v\} \\ u \in A_i, v \in B_i}} w_G(u, v) \le \text{vol}_G(A_i \cup B_i).$$

Thus, we have by (B.6) and the assumption of $\bar{\rho}(k) \ge \frac{1}{\log(n)}$ that

$$\mathbf{P}\left[|X - \mathbf{E}[X]| \ge c \cdot \mathbf{E}[X]\right] \le \frac{2(2 - \lambda_{n-k})}{c^2 \cdot C \cdot \log^3 n \cdot \bar{\rho}(k)^2}$$

$$= O\left(\frac{1}{\log n}\right).$$

Hence, by choosing a sufficient large constant $c$ and the union bound, we have that

$$w_H(A_i, B_i) = \Omega\left(w_G(A_i, B_i)\right) \text{ for all } 1 \leq i \leq k. \tag{B.7}$$

Next, we show that the degree of every vertex in $H$ is approximately preserved with high probability. Based on the random variable $Y_e$ defined in (B.2), we define the random variable $Z_u$ by

$$Z_u = \sum_{e:v\sim u} Y_e.$$

Then, the expected value of $Z_u$ is given by

$$\mathbf{E}[Z_u] = \sum_{e:v\sim u} \mathbf{E}[Y_e] = \sum_{e:v\sim u} p_e \cdot \frac{w_G(u,v)}{p_e}$$
$$= \sum_{v:v\sim u} w_G(u,v) = d_G(u),$$

and the second moment can be upper bounded by

$$\sum_{e:v\sim u} \mathbf{E}\left[Y_e^2\right] = \sum_{e:v\sim u} p_e \cdot \left(\frac{w_G(u,v)}{p_e}\right)^2$$
$$= \sum_{e:v\sim u} \frac{w_G(u,v)^2}{p_e} \leq \sum_{v:v\sim u} \frac{w_G(u,v)^2}{p_u(v)},$$

since $p_e \geq p_u(v)$. Now using the value of $p_u(v)$ from (3.1), we have

$$\sum_{e:v\sim u} \mathbf{E}\left[Y_e^2\right] \leq \sum_{v:v\sim u} w(u,v)^2 \cdot \frac{d_G(u) \cdot (2 - \lambda_{n-k})}{w(u,v) \cdot C \cdot \log^3 n}$$
$$= \frac{d_G(u) \cdot (2 - \lambda_{n-k})}{C \cdot \log^3 n} \sum_{v:v\sim u} w_G(u,v)$$
$$= \frac{d_G^2(u) \cdot (2 - \lambda_{n-k})}{C \cdot \log^3 n}$$

and for any edge $e = \{u,v\}$ we have that

$$0 \leq \frac{w(u,v)}{p_e} \leq \frac{w(u,v)}{p_u(v)} \leq \frac{d_G(u) \cdot (2 - \lambda_{n-k})}{C \cdot \log^3 n}.$$

Now, applying Bernstein's inequality (Lemma 12), we have

$$\mathbf{P}\left[|d_H(u) - d_G(u)| \geq \frac{d_u}{2}\right]$$
$$= \mathbf{P}\left[|Z_u - E[Z_u]| \geq \frac{\mathbf{E}[Z_u]}{2}\right]$$
$$\leq 2 \cdot \exp\left(\frac{-\frac{1}{8} \cdot d_G^2(u)}{\frac{d_G^2(u) \cdot (2-\lambda_{n-k})}{C \cdot \log^3 n} + \frac{1}{6} \cdot \frac{d_G^2(u) \cdot (2-\lambda_{n-k})}{C \cdot \log^3 n}}\right)$$
$$= 2 \cdot \exp\left(-\frac{\frac{1}{8} \cdot C \cdot \log^3 n}{\frac{7}{6} \cdot (2 - \lambda_{n-k})}\right)$$
$$= o\left(\frac{1}{n^2}\right).$$

Hence, it holds by the union bound that, with high probability, the degree of all the vertices in $H$ are approximately preserved up to a constant factor. This implies that for any subset $S \subseteq V$, we have

$$\mathrm{vol}_H(S) = \Theta\left(\mathrm{vol}_G(S)\right),$$

more specifically,

$$\mathrm{vol}_H(A_i \cup B_i) = \Theta\left(\mathrm{vol}_G(A_i \cup B_i)\right), \tag{B.8}$$

for all $1 \leq i \leq k$. Thus, combining (B.7) and (B.8) gives us that

$$\overline{\phi}_H(A_i, B_i) = \Omega\left(\overline{\phi}_G(A_i, B_i)\right) \tag{B.9}$$

for all $1 \leq i \leq k$, which implies that

$$\overline{\rho}_H(k) \geq \min_{1 \leq i \leq k} \overline{\phi}_H(A_i, B_i) = \min_{1 \leq i \leq k} \Omega\left(\overline{\phi}_G(A_i, B_i)\right)$$
$$= \Omega\left(\overline{\rho}_G(k)\right),$$

where the last equality follows from the fact that $\{(A_i, B_i)\}_{i=1}^k$ is the optimal cluster where $\overline{\rho}(k)$ is attained for graph $G$.

Next, we show that the top $(n-k)$-eigenspaces of $\mathcal{J}_G$ are preserved in $H$. Without loss of generality we assume the graph is connected. Since $\mathcal{J}_G = 2I - \mathcal{L}_G$ by definition, it holds that

$$\lambda_i(\mathcal{J}_G) = 2 - \lambda_{n+1-i}(\mathcal{L}_G). \tag{B.10}$$

Let

$$\mathcal{P} \triangleq \sum_{i=1}^{n-k}(2 - \lambda_i(\mathcal{L}_G))f_i f_i^{\mathsf{T}},$$

and with slight abuse of notation we call $\mathcal{P}^{-1/2}$ as the square root of the pseudo-inverse of $\mathcal{P}$, i.e.,

$$\mathcal{P}^{-1/2} = \sum_{i=1}^{n-k}(2 - \lambda_i(\mathcal{L}_G))^{-1/2} f_i f_i^{\mathsf{T}}.$$

Let $\overline{\mathcal{P}}$ be the projection on the span of $\{f_1, f_2, \cdots, f_{n-k}\}$, then

$$\overline{\mathcal{P}} = \sum_{i=1}^{n-k} f_i f_i^{\mathsf{T}}.$$

Recall that, for each vertex $v$, the indicator vector $\chi_v \in \mathbb{R}^n$ is defined by $\chi_v(u) = \frac{1}{\sqrt{d_G(v)}}$ if $u = v$ and $\chi_v(u) = 0$ otherwise. For each edge $e = \{u, v\}$ of $G$ we define a vector $g_e = \chi_u + \chi_v \in \mathbb{R}^n$ and a random matrix $X_e \in \mathbb{R}^{n \times n}$ by

$$X_e = \begin{cases} w_H(u, v) \cdot \mathcal{P}^{-1/2} g_e g_e^{\mathsf{T}} \mathcal{P}^{-1/2} & \text{if } e = \{u, v\} \text{ is sampled} \\ & \text{by the algorithm,} \\ \mathbf{0} & \text{otherwise.} \end{cases} \tag{B.11}$$

Then, it holds that

$$\sum_{e \in E} X_e = \sum_{e=\{u,v\} \in F} w_H(u, v) \cdot \mathcal{P}^{-1/2} g_e g_e^{\mathsf{T}} \mathcal{P}^{-1/2}$$
$$= \mathcal{P}^{-1/2} \left( \sum_{e=\{u,v\} \in F} w_H(u, v) \cdot g_e g_e^{\mathsf{T}} \right) \mathcal{P}^{-1/2}$$
$$= \mathcal{P}^{-1/2} \mathcal{J}_H' \mathcal{P}^{-1/2},$$

where

$$\mathcal{J}'_H \triangleq \sum_{e=\{u,v\}\in F} w_H(u,v) \cdot g_e g_e^\mathsf{T}$$

is the signless Laplacian matrix of $H$ normalised with respect to the degree of the vertices in the original graph $G$. We will now prove that, with high probability the top $n-k$ eigenspaces of $\mathcal{J}'_H$ and $\mathcal{J}_G$ are approximately the same. We first analyse the expectation of $\sum_{e\in E} X_e$, and have that

$$\mathbf{E}\left[\sum_{e\in E} X_e\right] = \sum_{e=\{u,v\}\in E} p_e \cdot w_H(u,v) \cdot \mathcal{P}^{-1/2} g_e g_e^\mathsf{T} \mathcal{P}^{-1/2}$$

$$= \sum_{e=\{u,v\}\in E} p_e \cdot \frac{w_G(u,v)}{p_e} \cdot \mathcal{P}^{-1/2} g_e g_e^\mathsf{T} \mathcal{P}^{-1/2}$$

$$= \mathcal{P}^{-1/2} \left(\sum_{e=\{u,v\}\in F} w_G(u,v) \cdot g_e g_e^\mathsf{T}\right) \mathcal{P}^{-1/2}$$

$$= \mathcal{P}^{-1/2} \mathcal{J}_G \mathcal{P}^{-1/2} = \sum_{i=1}^{n-k} f_i f_i^\mathsf{T} = \overline{\mathcal{P}}.$$

Moreover, for any edge $e=\{u,v\}\in E$ sampled by the algorithm, we have

$$\|X_e\| \le w_H(u,v) \cdot g_e^\mathsf{T} \mathcal{P}^{-1/2} \mathcal{P}^{-1/2} g_e = \frac{w_G(u,v)}{p_e} \cdot g_e^\mathsf{T} \mathcal{P}^{-1} g_e$$

$$\le \frac{w_G(u,v)}{p_e} \cdot \frac{1}{2-\lambda_{n-k}} \cdot \|g_e\|^2$$

$$\le \frac{2 w_G(u,v)}{p_u(v) + p_v(u)} \cdot \frac{1}{2-\lambda_{n-k}} \cdot \left(\frac{1}{d_G(u)} + \frac{1}{d_G(v)}\right)$$

$$\le \frac{2}{C \cdot \log^3 n},$$

where the second inequality follows by the min-max theorem of eigenvalues. Now we apply the matrix Chernoff bound (Lemma 13) to analyze the eigenvalues of $\sum_{e\in E} X_e$. Following Lemma 13 we set the parameters as follows:

$$\mu_{\max} = \lambda_{\max}\left(\mathbf{E}\left[\sum_{e\in E} X_e\right]\right) = \lambda_{\max}\left(\overline{\mathcal{P}}\right) = 1,$$

$$R = \frac{2}{C \cdot \log^3 n}, \text{ and} \tag{B.12}$$

$$\delta = \frac{1}{2}.$$

Then using the Matrix Chernoff bound (Lemma 13), we have

$$\mathbf{P}\left[\lambda_{\max}\left(\sum_{e\in E} X_e\right) \ge \frac{3}{2}\right] \le n \cdot \left(\frac{e^{\frac{1}{2}}}{1.5^{\frac{3}{2}}}\right)^{\frac{C\cdot\log^3 n}{2}} = O\left(\frac{1}{n^3}\right),$$

for some constant $C$; this implies that

$$\mathbf{P}\left[\lambda_{\max}\left(\sum_{e\in E} X_e\right) \le \frac{3}{2}\right] = 1 - O\left(\frac{1}{n^3}\right). \tag{B.13}$$

On the other hand, since $\mathbf{E}\left[\sum_{e\in E} X_e\right] = \overline{\mathcal{P}}$, we have $\mu_{\min} = 1$ and hence keeping $R$ and $\delta$ the same as above, using the Matrix Chernoff bound (Lemma 13), we get

$$\mathbf{P}\left[\lambda_{\min}\left(\sum_{e\in E} X_e\right) \le \frac{1}{2}\right] \le n \cdot \left(\frac{e^{-\frac{1}{2}}}{0.5^{\frac{1}{2}}}\right)^{\frac{C\cdot\log^3 n}{2}} = O\left(\frac{1}{n^3}\right);$$

this implies that

$$\mathbf{P}\left[\lambda_{\min}\left(\sum_{e\in E}X_e\right)\geq\frac{1}{2}\right]=1-O\left(\frac{1}{n^3}\right). \tag{B.14}$$

Combining (B.13), (B.14) and the fact that $\sum_{e\in E}X_e=\mathcal{P}^{-1/2}\mathcal{J}_H'\mathcal{P}^{-1/2}$, with probability $1-O\left(\frac{1}{n^3}\right)$ it holds for any non-zero $x\in\mathbb{R}^n$ in span$\{f_1,f_2,\cdots,f_{n-k}\}$ that

$$\frac{x^\intercal\mathcal{P}^{-1/2}\mathcal{J}_H'\mathcal{P}^{-1/2}x}{x^\intercal x}\in\left[\frac{1}{2},\frac{3}{2}\right]. \tag{B.15}$$

Let $y=\mathcal{P}^{-1/2}x$, and we rewrite (B.15) as

$$\frac{y^\intercal\mathcal{J}_H'y}{y^\intercal\mathcal{P}y}=\frac{y^\intercal\mathcal{J}_H'y}{y^\intercal y}\cdot\frac{y^\intercal y}{y^\intercal\mathcal{P}y}\in\left[\frac{1}{2},\frac{3}{2}\right].$$

Since $\dim(\text{span}\{f_1,f_2,\cdots,f_{n-k}\})=n-k$, there exist $n-k$ orthogonal vectors whose Rayleigh quotient with respect to $\mathcal{J}_H'$ is $\Theta(\lambda_{n-k}(2I-\mathcal{L}_G))$. Hence, by the Courant-Fischer Theorem (Theorem 11) we have

$$\frac{1}{2}\cdot\lambda_{n-k}(2I-\mathcal{L}_G)\leq\lambda_{k+1}(\mathcal{J}_H')\leq\frac{3}{2}\cdot\lambda_{n-k}(2I-\mathcal{L}_G) \tag{B.16}$$

By the definition of $\mathcal{J}_H'=D_G^{-1/2}\left(D_H+A_H\right)D_G^{-1/2}$, we have

$$\begin{aligned}\mathcal{J}_H&=D_H^{-1/2}\left(D_H+A_H\right)D_H^{-1/2}\\&=D_H^{-1/2}\left(D_G^{1/2}\cdot\mathcal{J}_H'\cdot D_G^{1/2}\right)D_H^{-1/2}.\end{aligned}$$

Hence, we set $y=D_G^{1/2}D_H^{-1/2}x$ for any $x\in\mathbb{R}^n$ and have that

$$\begin{aligned}\frac{x^\intercal\mathcal{J}_Hx}{x^\intercal\cdot x}&=\frac{x^\intercal\cdot D_H^{-1/2}\left(D_G^{1/2}\cdot\mathcal{J}_H'\cdot D_G^{1/2}\right)D_H^{-1/2}\cdot x}{x^\intercal\cdot x}\\&=\frac{y^\intercal\cdot\mathcal{J}_H'\cdot y}{x^\intercal\cdot x}\geq\frac{1}{2}\cdot\frac{y^\intercal\cdot\mathcal{J}_H'\cdot y}{y^\intercal\cdot y},\end{aligned} \tag{B.17}$$

where we use the fact that the degree of a vertex differs by a constant factor between $H$ and $G$. Similarly, we also have

$$\frac{x^\intercal\cdot\mathcal{J}_H\cdot x}{x^\intercal\cdot x}\leq\frac{3}{2}\cdot\frac{y^\intercal\cdot\mathcal{J}_H'\cdot y}{y^\intercal\cdot y}, \tag{B.18}$$

Let $T\subset\mathbb{R}^n$ be a $(k+1)$-dimensional subspace of $\mathbb{R}^n$ satisfying

$$\lambda_{k+1}(\mathcal{J}_H)=\max_{x\neq0,x\in T}\frac{x^\intercal\cdot\mathcal{J}_H\cdot x}{x^\intercal\cdot x},$$

and $\widetilde{T}=\left\{D_G^{1/2}D_H^{-1/2}x:x\in T\right\}$. Since $D_G^{1/2}D_H^{-1/2}$ has full rank, $\widetilde{T}$ is also a $(k+1)$-dimensional subspace of $\mathbb{R}^n$. Hence, by the Courant-Fischer Theorem (Theorem 11) and (B.17), we have that

$$\begin{aligned}\lambda_{k+1}(\mathcal{J}_H')&=\min_{\substack{S\\\dim(S)=k+1}}\max_{y\in S\setminus\{\mathbf{0}\}}\frac{y^\intercal\cdot\mathcal{J}_H'\cdot y}{y^\intercal\cdot y}\\&\leq\max_{y\in\widetilde{T}\setminus\{\mathbf{0}\}}\frac{y^\intercal\cdot\mathcal{J}_H'\cdot y}{y^\intercal\cdot y}\\&\leq2\cdot\max_{x\in T\setminus\{\mathbf{0}\}}\frac{x^\intercal\cdot\mathcal{J}_H\cdot x}{x^\intercal\cdot x}=2\cdot\lambda_{k+1}(\mathcal{J}_H).\end{aligned} \tag{B.19}$$

Next, using (B.16) and (B.19), we have

$$\frac{1}{2} \cdot \lambda_{k+1}(\mathcal{J}_G) \leq \lambda_{k+1}(\mathcal{J}_H') \leq 2 \cdot \lambda_{k+1}(\mathcal{J}_H),$$

which implies that

$$\frac{1}{4} \cdot \lambda_{k+1}(\mathcal{J}_G) \leq \lambda_{k+1}(\mathcal{J}_H). \tag{B.20}$$

Similarly, let $U \subset \mathbb{R}^n$ be an $(n-k)$-dimensional subspace of $\mathbb{R}^n$ satisfying

$$\lambda_{k+1}(\mathcal{J}_H) = \min_{x \neq 0, x \in U} \frac{x^\mathsf{T} \cdot \mathcal{J}_H \cdot x}{x^\mathsf{T} \cdot x},$$

and $\widetilde{U} = \left\{ D_G^{1/2} D_H^{-1/2} x : x \in U \right\}$. Since $D_G^{1/2} \cdot D_H^{-1/2}$ has full rank, $\widetilde{U}$ is also an $(n-k)$-dimensional subspace of $\mathbb{R}^n$. Thus, using the Courant-Fischer Theorem (Theorem 11) and (B.18), we have

$$\begin{aligned}
\lambda_{k+1}(\mathcal{J}_H') &= \max_{\substack{S \\ \dim(S) = n-k}} \min_{y \in S \setminus \{\mathbf{0}\}} \frac{y^\mathsf{T} \cdot \mathcal{J}_H' \cdot y}{y^\mathsf{T} \cdot y} \\
&\geq \min_{y \in \widetilde{U} \setminus \{\mathbf{0}\}} \frac{y^\mathsf{T} \cdot \mathcal{J}_H' \cdot y}{y^\mathsf{T} \cdot y} \\
&\geq \frac{2}{3} \cdot \min_{x \in U \setminus \{\mathbf{0}\}} \frac{x^\mathsf{T} \cdot (2I - \mathcal{L}_H) \cdot x}{x^\mathsf{T} \cdot x} \\
&= \frac{2}{3} \cdot \lambda_{k+1}(\mathcal{J}_H).
\end{aligned} \tag{B.21}$$

Next, by (B.16) and (B.21) we have

$$\frac{2}{3} \cdot \lambda_{k+1}(\mathcal{J}_H) \leq \gamma_{k+1}(\mathcal{L}_H') \leq \frac{3}{2} \cdot \lambda_{k+1}(\mathcal{J}_G),$$

which implies that

$$\lambda_{k+1}(\mathcal{J}_H) \leq \frac{9}{4} \cdot \lambda_{k+1}(\mathcal{J}_G). \tag{B.22}$$

Thus, combining (B.20) and (B.22) we have

$$\frac{1}{4} \cdot \lambda_{k+1}(\mathcal{J}_G) \leq \lambda_{k+1}(\mathcal{J}_H) \leq \frac{9}{4} \cdot \lambda_{k+1}(\mathcal{J}_G),$$

Hence, the the top $n-k$ eigenspaces of $\mathcal{J}_G$ are preserved in $\mathcal{J}_H$. This proves the second statement of the theorem.

## C. Omitted Detail from Section 4

In this section we list all the proofs omitted from Section 4.

*Proof of Lemma 7.* The proof follows from (Macgregor & Sun, 2021a), which proves the result for undirected graphs. We include the proof here for completeness. Let $S = A_1 \cup B_2$ in $H$, then

$$\begin{aligned}
\phi_H(A_1 \cup B_2) = \phi_H(S) &= \frac{w_H(S, V \setminus S)}{\mathrm{vol}_H(S)} \\
&= \frac{\mathrm{vol}_H(S) - 2w_H(S, S)}{\mathrm{vol}_H(S)} \\
&= 1 - \frac{2w_H(S, S)}{\mathrm{vol}_H(S)} \\
&= 1 - \frac{2w_{\overrightarrow{G}}(A, B)}{\mathrm{vol}_{\mathrm{out}}(A) + \mathrm{vol}_{\mathrm{in}}(B)} \\
&= f_{\overrightarrow{G}}(A, B).
\end{aligned} \tag{C.1}$$

This proves the first statement of the lemma. The second statement of the lemma follows by the similar argument. $\square$

*Proof of Lemma 8.* By definition, we have that

$$f_{\overrightarrow{G}}(A, B) = 1 - \overline{\phi}_{\overrightarrow{G}}(A, B), \tag{C.2}$$

and this implies that

$$
\begin{aligned}
\bar{\rho}_{\overrightarrow{G}}(k) &= \max_{(A_1,B_1),\dots,(A_k,B_k)} \min_{1 \le i \le k} \overline{\phi}_{\overrightarrow{G}}(A_i, B_i) \\
&= \max_{(A_1,B_1),\dots,(A_k,B_k)} \min_{1 \le i \le k} \left(1 - f_{\overrightarrow{G}}(A_i, B_i)\right) \\
&= 1 - \min_{(A_1,B_1),\dots,(A_k,B_k)} \max_{1 \le i \le k} f_{\overrightarrow{G}}(A_i, B_i) \\
&= 1 - \min_{C_1,\dots,C_k} \max_{1 \le i \le k} \phi_H(C_i),
\end{aligned}
$$

where the second line follow by (C.2), and the last one follows by Lemma 7 and $C_i = A_{i_1} \cup B_{i_2}$. $\qquad\square$

