# OpenReview forum: "Online Sparsification of  Bipartite-Like Clusters in Graphs"
_ICML.cc/2025/Conference — ICML 2025 poster_

### Official Review · Reviewer_7SXC · 2025-03-09

**Overall Recommendation:** 4

**Summary:**

The paper studies graph sparsifiers that preserve bipartite-like communities in undirected and directed graphs. The notion of communities is formalized via the bipartiteness ratio and then generalized to $k$ communities in the standard way through $k$-way partitions.

For undirected graphs, the novel sparsifier is obtained by subsampling from the original graph using a technique similar to one proposed by Sun and Zanetti (2019).

For directed graphs, the construction is more involved and first constructs an undirected graph which can be sparsified using a result by Sun and Zanetti, and then the undirected sparsified graph can be made directed again.

**Claims And Evidence:**

There is good evidence for the theoretical claims, including full proofs in the appendix of the submission.

**Essential References Not Discussed:**

Not to my knowledge.

**Experimental Designs Or Analyses:**

- The experiments should contain a detailed discussion how the sampling parameters were set. This seems to be missing at this point. This is particularly important since the theoretical sampling probabilities contain terms like $\lambda_{n-k}$ and it is not obvious how this will be computed (I guess using power iteration? But then what do you do for local algorithms – see also next point.).
- I was surprised that for both undirected and directed graphs only a single algorithm was used for the evaluation of the sparsifier. Additionally, I was surprised that only a local algorithm was used when evaluating the sparsifier for undirected graphs.
- The graphs used in the experiments are relatively small, never containing more than 5000 nodes for the synthetic datasets. For the real-world datasets I am not sure about their exact size, but the running times suggest that they are also rather small.

**Methods And Evaluation Criteria:**

Generally they make sense, but I think the experiments could be more comprehensive.

**Other Comments Or Suggestions:**

- In Theorems 1 and 2, there are no bounds on the size of the sparsifier. In some sense, they are implicit in the running time, but I think they should be made explicit.
- I would appreciate a more comprehensive experimental evaluation on more and larger real-world datasets.
- Theorem 5 is missing a statement about the running time for constructing the sparsifier.
- Theorem 2 appears to be a bit informal and a more formal version of it (similar to Theorem 5 as formal version of Theorem 1) would be a good addition.

**Update after rebuttal:** The authors have given a thorough reply to my comments and I have thus increased my score. I will appreciate when the authors discuss the choice and impact of the sampling probability thoroughly in their experiments.

**Other Strengths And Weaknesses:**

The biggest strength of the paper are the theoretical results which are interesting and non-trivial to obtain. The paper could be improved with a better experimental evaluation, though. Overall, I think the positives outweigh the negatives, given that this is mostly a theory paper.

**Questions For Authors:**

- How did you set the sampling probabilities in the experiments? How did you deal with the term $\lambda_{n-k}$, especially for the local algorithms.
- For the Lithuania dataset it seems like the gap in bipartiteness ratio between the two algorithms is relatively large. Do you have a justification for this?

**Relation To Broader Scientific Literature:**

I think the key references are cited.

**Theoretical Claims:**

I did not check the proofs in detail.

---

> ### Author Rebuttal · Authors · 2025-03-31
>
> We thank the reviewer for their positive evaluation and detailed comments. Here is our response to the raised questions:
>
> **Response to _Other Comments Or Suggestions_:**
>
> > In Theorems 1 and 2, there are no bounds on the size of the sparsifier. In some sense, they are implicit in the running time, but I think they should be made explicit.
>
> In the proof of Theorem 3, we proved on Lines 623 of the appendix that the number of edges in our sparsifier is
> $$
> O\left(\frac{n\log^3n}{2-\lambda_{n-k}}\right),
> $$
> and the number of edges in our constructed directed sparsifier is dominated by the term above as well. We'll make it clear in the next version of the paper.
>
> > I would appreciate a more comprehensive experimental evaluation on more and larger real-world datasets.
>
> We agree that a more comprehensive experimental evaluation on our developed algorithms will make our paper considerably stronger. We'll add more experimental results in the next version of the paper.
>
> > Theorem 5 is missing a statement about the running time for constructing the sparsifier.
>
> The algorithm behind Theorem 5 runs in nearly-linear time in the number of edges of the input graph. We will make it clear in the next version of the paper.
>
> > Theorem 2 appears to be a bit informal and a more formal version of it (similar to Theorem 5 as formal version of Theorem 1) would be a good addition.
>
> In the next version, we will add a more formal version of Theorem 2, and add it to Section 4. The role of the more formal theorem and Theorem 2 will be similar with Theorem 5 and Theorem 1.
>
> **Response to _Questions For Authors_:**
>
> > How did you set the sampling probabilities in the experiments? How did you deal with the term $\lambda_{n-k}$, especially for the local algorithms.
>
> Our sampling probability is defined in Equation (3.1) of the submission. By the assumption of the $k$-way expansion and the higher-order dual-Cheeger inequality, we treat $C\log^3(n)/(2-\lambda_{n-k})$ as $O(\log^c(n))$ for a constant $c$. This will only influence the total number of sampled edges and the algorithm's overall running time by a factor of $\log^c(n)$. We employed this trick in our experiment, and will state this in detail in the next version of the paper.
>
> > For the Lithuania dataset it seems like the gap in bipartiteness ratio between the two algorithms is relatively large. Do you have a justification for this?
>
> This is a very interesting observation. We think this dues to the fact that, while we apply different seed vertices as the input of the local algorithm and the algorithm returns the output clusters of potentially very different sizes, we employ the same sampling probability to construct a sparsifier. As such, the bipartiteness ratios returned by our algorithm and the MS algorithm could scale differently.
>
> We would like to thank the reviewer once more for these insightful comments.  We will implement these suggestions in the next version of the paper.

---

### Official Review · Reviewer_26Yv · 2025-03-10

**Overall Recommendation:** 2

**Summary:**

In this paper, the authors study study bipartite-like clusters and present efficient and online algorithms that find such clusters in both undirected graphs and directed ones. Experiments on real and synthetic graphs demonstrates that the proposed algorithm can speedup the existing algorithm.

**Claims And Evidence:**

The title of the paper is "Finding Bipartite-like Clusters on the Fly", however, the proposed techniques seems like sparsifier regarding bipartite-like clustering.  Amore appropriate title may be better.

**Essential References Not Discussed:**

No

**Experimental Designs Or Analyses:**

1. For the synthetic datasets, it is more convincing if more generation models can be used to generate synthetic datasets.
2. The size of the datasets used in the paper is unclear.

**Methods And Evaluation Criteria:**

1.  The fields of the datasets used in the experiments is limited. It seems more convincing if more datasets from various areas can be used to evaluate the proposed methods.

**Other Comments Or Suggestions:**

No

**Other Strengths And Weaknesses:**

Weakness:

W1. The significance to find the bipartite-like clusters in a graph seems weak. It is more convincing if more direct application scenerios can be provided.

**Questions For Authors:**

See above

**Relation To Broader Scientific Literature:**

The proposed method can be used to speedup finding bipartite-like clusters.

**Theoretical Claims:**

Yes

---

> ### Author Rebuttal · Authors · 2025-03-30
>
> We thank the reviewer's work and the report. Here is our response to the raised questions:
>
> **Response to _Claims And Evidence_**:
>
> >The title of the paper is "Finding Bipartite-like Clusters on the Fly", however, the proposed techniques seems like sparsifier regarding bipartite-like clustering. A more appropriate title may be better.
>
> We thank the reviewer for the comment. In the next version of the paper, we will update the title to reflect their suggestion.
>
> **Response to _Experimental Designs Or Analyses_**:
>
> >For the synthetic datasets, it is more convincing if more generation models can be used to generate synthetic datasets.
>
> Since SBMs are commonly used models to generate synthetic graphs for clustering algorithms, it's not very clear to us which other generation models the reviewer refers to. If the reviewer has a specific generation model in mind that is suitable for our problem, we'll be happy to add additional experiments to make our work more convincing.
>
> >The size of the datasets used in the paper is unclear.
>
> The datasets used in our work are the common ones studying the problem, and are the ones used in the previous work, e.g., [3]. Following the reviewer's comments, we will include the size of the datasets in the next version of the paper.
>
> **Response to _Other Strengths And Weaknesses_:**
>
> >W1. The significance to find the bipartite-like clusters in a graph seems weak. It is more convincing if more direct application scenarios can be provided.
>
> We respectfully disagree with the reviewer's comment on the _significance_ of finding bipartite-like clusters in a graph due to the following reasons:
>
> First of all, finding bipartite-like clusters in a graph is one of the most important problems in theoretical computer science, and is a natural generalisation of the max cut problem; recall that the max cut problem is closely linked to the unique games conjecture. In particular, Trevisan [1] proved that algorithms for finding bipartite-like clusters can be employed to design an approximation algorithm for the max cut problem, and this approach has remained the only combinatorial one for designing approximation algorithms for the max cut problem.
>
> Secondly, the algorithms and their complexities for finding bipartite-like clusters in a graph relate to understanding the top eigenspace of a graph's normalised Laplacian matrix, and this problem is studied separately in spectral graph theory community [2]. This is another indication for its importance.
>
> Thirdly, the problem of finding bipartite-like clusters in a graph has been also actively investigated in the machine learning community. For instance, Macgregor and Sun [3] presented local algorithms for finding bipartite-like clusters; the significance of their work is clearly recognised by an ICML'21 oral presentation. With respect to more applied domains, our problem is closely linked to finding clusters in disassortative networks and has been applied in training neural networks, e.g., learning over networks with heterophily [4].
>
> We hope that the above-mentioned research fields, in which finding bipartite-like clusters has been studied, could convince the reviewer on the significance of the problem. Building on this sequence of research over the past 15 years, our work presents the first algorithms that sparsify the input instances of both the undirected and directed graphs. Hence, we feel that our work is important from this perspective. In the next version of the paper, we will expand the section of related works to better highlight the significance of the problem.
>
> _Additional Comments_:
>
> We hope that we have answered all of the reviewer's concerns on our work. Given that the reviewer's overall evaluation is mainly due to its significance (instead of flawed proofs or easily obtained theoretical results), we hope that the reviewer can take our response and other reviewers' reports into account when further evaluating our submission. Many thanks.
>
> **Reference**
>
> 1. Trevisan, L. Max cut and the smallest eigenvalue. STOC'09.
>
> 2. Liu, S. Multi-way dual Cheeger constants and spectral bounds of graphs. Advances in Mathematics, 2015.
>
> 3. Macgregor, P. and Sun, H. Local algorithms for finding densely connected clusters. ICML'21.
>
> 4. Zhu, J., Yan, Y., Zhao, L., Heimann, M., Akoglu, L., and Koutra, D. Beyond homophily in graph neural networks: Current limitations and effective designs. NeurIPS'20

---

### Official Review · Reviewer_s5UQ · 2025-03-13

**Overall Recommendation:** 3

**Summary:**

The paper propose efficient and online algorithms to detect bipartite-like clusters for both directed and undirected graphs. They both are graph sparsifiers that can sparsify the graph into $\tilde{O}(n)$ edges while preserving the bipartite clusters with high probability.

**Claims And Evidence:**

The claims are well supported

**Essential References Not Discussed:**

The paper "Algorithmic Tools for Understanding the Motif Structure of Networks" in ECML 2022 shows by finding square-dense and triangle-sparse subgraphs can lead to bipartite-like subgraphs, and leveraged this idea to detect anomalies in social networks.

**Experimental Designs Or Analyses:**

The speedup is not obvious in figure 3, and the running time fluctuates around 1750 nodes. Some explanations here could be helpful.

**Methods And Evaluation Criteria:**

The evaluation methods make sense.

**Other Comments Or Suggestions:**

What if $A_i$ and $B_i$ are not required to be disjoint? In some directed graph problems the "sender" and "receiver" can be considered as different roles.

**Other Strengths And Weaknesses:**

The problem of detecting communities as bipartite-like structures in graphs is not well-motivated in the paper.

Authors also should highlight that the proposed conductance is not only find bipartite-like structure within the cluster, but also penalizing the connections from the cluster to outside.

**Questions For Authors:**

N.A.

**Relation To Broader Scientific Literature:**

The algorithms proposed is highly related to the cluster-preserved sparsifier work from Sun and Zanetti 2019.

**Theoretical Claims:**

The claims are well supported

---

> ### Author Rebuttal · Authors · 2025-03-31
>
> We thank the reviewer for their positive evaluation and detailed comments. Here is our response to the raised questions:
>
> **Response to _Experimental Designs Or Analyses_**:
>
> >The speedup is not obvious in figure 3, and the running time fluctuates around 1750 nodes. Some explanations here could be helpful.
>
> This is a very interesting and meaningful question. Notice that the sampling probability used in our algorithm is defined in Equation (3.1) of the submission. By the assumption of the $k$-way expansion and the higher-order dual-Cheeger inequality, we always set  $C\log^3(n)/(2-\lambda_{n-k})$, the quantity involved in the sampling probability, as a fixed constant in the experiment. On the other hand,  by (3.1) the sampling probability of every edge doesn't change linearly with respect to the linear increase of $n$. We believe this is the reason behind the fluctuation shown in Figure 3. We will add necessary explanation in the next version of the paper.
>
> **Response to _Essential References Not Discussed_**:
>
> >The paper "Algorithmic Tools for Understanding the Motif Structure of Networks" in ECML 2022 shows by finding square-dense and triangle-sparse subgraphs can lead to bipartite-like subgraphs, and leveraged this idea to detect anomalies in social networks.
>
> Thank you for pointing out this reference. Indeed, this paper is closely related to our submission. In the next version of our paper, we will add necessary discussions of this ECML'22 paper and other related works.
>
> **Response to _Other Strengths And Weaknesses_**:
>
> >The problem of detecting communities as bipartite-like structures in graphs is not well-motivated in the paper.
>
> In the next version, we will expand our Introduction section, and add the following discussion to better motivate our studied problem:
>
> First of all, finding bipartite-like structures in a graph is one of the most important problems in theoretical computer science, and is a natural generalisation of the max cut problem; recall that the max cut problem is closely linked to the unique games conjecture. In particular, Trevisan [1] proved that algorithms for finding bipartite-like clusters can be employed to design an approximation algorithm for the max cut problem, and this approach has remained the only combinatorial one for designing approximation algorithms for the max cut problem.
>
> Secondly, the algorithms and their complexities for finding bipartite-like clusters in a graph relate to understanding the top eigenspace of a graph's normalised Laplacian matrix, and this problem is studied separately in spectral graph theory community [2]. This is another indication for its importance.
>
> Thirdly, the problem of finding bipartite-like structures in a graph has been also actively investigated in the machine learning community. For instance, Macgregor and Sun [3] presented local algorithms for finding bipartite-like clusters. With respect to more applied domain, our problem is closely linked to finding clusters in disassortative networks and has been applied in training neural networks, e.g., learning over networks with heterophily [4].
>
>
> >Authors also should highlight that the proposed conductance is not only find bipartite-like structure within the cluster, but also penalizing the connections from the cluster to outside.
>
> Thanks a lot for pointing out this. This is an excellent suggestion, and we will better highlight this point in the next version of our paper.
>
> **Response to _Other Comments Or Suggestions_**:
>
> > What if $A_i$ and $B_i$ are not required to be disjoint? In some directed graph problems the "sender" and "receiver" can be considered as different roles.
>
> It is a really interesting question, and it seems that our current technique cannot be easily adjusted to handle this situation. We believe that this could be a meaningful question for future work.
>
> We would like to thank the reviewer once more for these valuable comments. We will implement these suggestions in the next version of the paper.
>
> **Reference**
>
> 1. Trevisan, L. Max cut and the smallest eigenvalue. STOC'09.
>
> 2. Liu, S. Multi-way dual Cheeger constants and spectral bounds of graphs. Advances in Mathematics, 2015.
>
> 3. Macgregor, P. and Sun, H. Local algorithms for finding densely connected clusters. ICML'21.
>
> 4. Zhu, J., Yan, Y., Zhao, L., Heimann, M., Akoglu, L., and Koutra, D. Beyond homophily in graph neural networks: Current limitations and effective designs. NeurIPS'20

---

### Official Review · Reviewer_Mk9R · 2025-03-15

**Overall Recommendation:** 4

**Summary:**

This paper studies the problem of finding bipartite-like clusters in both directed and undirected graphs. The authors propose a novel graph sparsification algorithm that can be implemented online and preserves the structure of bipartite-like clusters.  The main findings are theoretical results proving that their algorithm, which runs in nearly-linear time, produces a sparse subgraph with $\tilde{O}(n)$ edges that maintains the key bipartite-like cluster properties of the original graph. The main algorithmic idea is to perform a specific type of edge sampling based on the dual Cheeger constant and degrees. For directed graphs, a reduction to an undirected "semi-double cover" graph is introduced before sparsification, and a "reverse semi-double cover" operation is used to transform the sparsified undirected graph back into a directed one.  The authors also demonstrate empirically that their algorithm significantly speeds up existing local clustering algorithms while preserving the quality of the results.

**Claims And Evidence:**

The main claims are well-supported by theoretical evidence (proofs of Theorems 1,2 and 5) and experimental evidence.

* Undirected Graphs:  The sparsified graph G* preserves the k-way dual Cheeger constant (and hence the bipartite-like cluster structure) and has $\tilde{O}(n)$ edges (Theorem 1/5). The proof is sketched in Section 3, while the remaining details are shown in Appendix B.

* Directed Graphs:  A similar result holds for directed graphs, using the semi-double cover construction (Theorem 2). The proof is sketched in Section 4, with the remaining details in the Appendix.

* Practical Speedup: The authors validate their theoretical findings with an empirical evaluation showcasing that the sparsification algorithm speeds up existing local clustering algorithms in practice.  The evidence is experimental results on both synthetic and real-world datasets.

The claims seem strong and supported by sound evidence.

**Essential References Not Discussed:**

I do not believe there are essential references that are not discussed.

**Experimental Designs Or Analyses:**

I reviewed the soundness of the experimental designs and analyses. The datasets and evaluation criteria follow that of (Macgregor & Sun, 2021a) and is reasonable.

The experimental design appears sound and the analyses are clearly presented.

**Methods And Evaluation Criteria:**

The proposed methods and evaluation criteria are appropriate.

* Sparsification: Using graph sparsification to accelerate clustering is a sensible approach.  The specific sampling scheme based on the dual Cheeger constant and node degrees is novel and theoretically sound.

* Semi-Double Cover: The reduction from directed to undirected graphs via the semi-double cover is a clever technique to leverage the undirected sparsification algorithm.  The reverse operation is well-defined.

* Empirical Evaluation: The evaluation uses standard metrics, derived from the definition of the problem, to do their experiments. The results are reported over a number of runs. Both synthetic and real-world datasets are used, similarity to (Macgregor & Sun, 2021a).

**Other Comments Or Suggestions:**

Consider explicitly mentioning what do you mean by online in the paper. Online as in dynamic algorithm, or online as in online algorithms (where decisions are irreversible )? The oracle with the degrees of the nodes is with respect to the current graph, or with respect to the graph after all node/edge insertions?

Page 2, line 60: "since most sparsification algorithms are only applicable for undirected graphs", did you mean directed graphs?

Explain better how do you use the SBM model to generate bipartite graphs. The way it is described it seems to be generating non-bipartite graphs.

Give some intuition on the bipartiteness ratio and the flow-ratio used in the experiments. If I understand this correctly, one wants to find clusters with lower bipartiteness (and flow-ratio), so this means that with your sparsification does better compared to running an algorithm on the initial graph?

**Other Strengths And Weaknesses:**

Strengths:

* Novelty: The proposed sparsification algorithm for bipartite-like clusters is novel and theoretically well-justified.

* Generality: The algorithm works for both undirected and directed graphs, which is a significant advantage. The reduction to the undirected case for directed graphs is clever.

* Efficiency: The algorithm is nearly-linear time, and can be implemented using a local algorithm, making it suitable for large graphs.

* Practical Impact:  The experimental results demonstrate significant speedups for existing local clustering algorithms.

* Clarity: The paper is well-written and clearly explains the problem, the proposed algorithm, and the theoretical and experimental results.

Weaknesses:

* Some concepts and metrics can be explained better (see questions).

**Questions For Authors:**

Could you elaborate more on the practical implications of the lower bound restriction on the dual Cheeger constant ($\overline{\rho}_G(k) \geq 1/ \log n$)? Are there specific types of graphs or applications where this condition is likely to be met or not met? How does the algorithm's performance degrade as $\overline{\rho}_G(k)$ approaches this lower bound?

Could you do the extra effort and add an experiment for graphs with a higher number of clusters in the SBM model? Given that the algorithm by (Macgregor & Sun, 2021a) stops once it finds a good cluster, doesn't this mean it could find subset of a cluster, that I assume you remove from the graph, and it can affect the subsequent clusters that it finds?

Could you explain why it doesn't make sense to compare to spectral sparsifiers in practice?

See also some questions in the comments/suggestions section.

**Relation To Broader Scientific Literature:**

The paper is well-situated within the broader scientific literature on graph clustering, graph sparsification, and spectral graph theory.  It cites relevant prior work on: Graph Clustering, Graph Sparsification, Spectral Graph Theory.

The paper clearly distinguishes its contributions from prior work. It emphasizes the novelty of preserving the inter-connection between vertex sets (bipartite-like clusters) rather than just the cut values between a set and its complement. It further outlines the importance of practical sparsification algorithms that can easily be implemented.

**Theoretical Claims:**

I checked the correctness of the proofs for the main theorems (Theorem 1/5 and Theorem 2) as best as possible. I followed the high-level arguments and checked the key steps in the proofs.

The proofs appear to be mathematically sound and well-structured, leveraging standard techniques appropriately.

---

> ### Author Rebuttal · Authors · 2025-03-30
>
> We thank the reviewer for their positive and such detailed report. Here is our response to the raised questions:
>
> **Response to _Other Comments Or Suggestions:_**
>
> >Consider explicitly mentioning what do you mean by online in the paper.
>
> Online in our setting is more relevant to online algorithms and, with the degree oracle for the whole graph, our algorithm decides whether to keep every online arriving edge in a sparsifier or not without the global information of the graph. We will make this clearer in the next version of the paper.
>
> >Page 2, line 60: "since most sparsification algorithms are only applicable for undirected graphs", did you mean directed graphs?
>
> No. We do mean "undirected graphs" in this specific place, and most sparsification algorithms are indeed only applicable for undirected graphs.
>
> >Explain better how do you use the SBM model to generate bipartite graphs. The way it is described it seems to be generating non-bipartite graphs.
>
> We explained the SBM model on Lines 354- 388 (left) for undirected graphs, and on Lines 374-383 (right) for directed graphs. The reviewer is right that indeed we use SBM to generate non-bipartite graphs. However, since vertices in the same cluster are connected with much lower probability, our generated graphs are _almost bipartite_. As shown from the experiments, our algorithms are able to find the two partitions of the clusters. Notice that the our task becomes trivial if the underlying input graph is a bipartite graph. In the next version of the paper we'll better explain this part.
>
> >Give some intuition on the bipartiteness ratio and the flow-ratio used in the experiments.
>
> Two clusters $A$ and $B$ with a low bipartiteness/flow ratio corresponds to the case that most edges leaving vertices in $A$ (resp. $B$) will go to $B$ (resp. $A$), and in comparison there are fewer edges inside $A$ and $B$. The objective for finding bipartite clusters is to approximately find the sets $A$ and $B$. We prove that this task can be achieved with easily implementable sparsification algorithms. Our developed algorithms can be used to speed up the running time of the overall algorithm framework for finding two clusters.
>
> **Response to _Questions For Authors:_**
>
> >Could you elaborate more on the practical implications of the lower bound restriction on the dual Cheeger constant ($\overline{\rho}_G(k)\geq 1/\log n)$)? Are there specific types of graphs or applications where this condition is likely to be met or not met? How does the algorithm's performance degrade as $\overline{\rho}_G(k)$ approaches this lower bound?
>
> A graph with $\overline{\rho}_G(k)\geq 1/\log n$ implies that $G$ has at least $k$ mutually disjoint $A_i$ and $B_i$ such that
> $$
> \frac{2w(A_i, B_i)}{\mathrm{vol}(A_i\cup B_i)} \geq \frac{1}{\log n};
> $$
> that is, most edges adjacent to vertices in $A_i\cup B_i$ are between $A_i$ and $B_i$, and $A_i$ and $B_i$ forms an _almost_ bipartite graph.
>
> In practice, this condition are usually met when we study $k$ pairs of vertex groups that are densely connected. Our experimental results on the Interstate Disputes Dataset and the Migration Dataset are two good examples.
>
> Our sampled edges are inversely proportional to $\overline{\rho}_G(k)$. The smaller the value of $\overline{\rho}_G(k)$, the more sampled edges needed in order to achieve the proved guarantee. We'll make it clear in the next version of the paper.
>
> >Could you do the extra effort and add an experiment for graphs with a higher number of clusters in the SBM model? Given that the algorithm by (Macgregor & Sun, 2021a) stops once it finds a good cluster, doesn't this mean it could find subset of a cluster, that I assume you remove from the graph, and it can affect the subsequent clusters that it finds?
>
> In the next version of the paper, we will add an experiment for graphs with more number of clusters. The algorithm by Macgregor & Sun is a local one. If we want to find $k$ pairs of clusters, several other algorithms can be used to achieve this objective.
>
> >Could you explain why it doesn't make sense to compare to spectral sparsifiers in practice?
>
> In our point of view, it doesn't make sense to compare our result to spectral sparsifiers for 3 reasons:
>
> 1. Our focus is to construct a sparsifier which preserves the cut values $w(A_i, B_i)$ for $k$ pairs of $A_i$ and $B_i$. A spectral sparsifier only preserves the cut values between any vertex set $A$ and _its complement $V\setminus A$_. Hence, a spectral sparsifier doesn't achieve our target;
>
> 2. Our algorithms work for both undirected and directed graphs, but a spectral sparsifier only works for undirected ones.
>
> 3. Our work is easy to implement, but most algorithms for constructing spectral sparsifiers are based on Laplacian solvers or complicated expander decomposition schemes making it difficult to implement.
>
> We have tried to address all of your questions within the word limit. We'll be happy to answer any other questions during the discussion phase.

---

### Decision · Program_Chairs · 2025-05-01

**Decision:**

Accept (poster)

**Comment:**

This paper addresses the problem of identifying bipartite-like clusters, where two sets of vertices are densely connected, moving beyond traditional conductance-based clustering. The authors present novel, nearly-linear time online sparsification algorithms for both undirected and directed graphs that preserve these bipartite-like cluster structures. Experimental results on synthetic and real-world data demonstrate the effectiveness and significant speedup achieved by incorporating these sparsifiers into existing clustering algorithms.

This paper presents a valuable contribution and would be a good addition to the ICML program. Its core strengths lie in the proposed sparsification algorithm for bipartite-like clusters is novel and theoretically well-justified, addressing an increasingly important aspect of graph clustering. A particularly strong point is that the algorithm works for both undirected and directed graphs, which is a significant advantage, broadening its applicability. Furthermore, the algorithm is nearly-linear time, making it practical for large datasets. A minor weakness is that it would be nice to have additional motivation on studying the problem on bipartite graphs; while examples are given, slightly expanding on the significance and application domains could further enhance the paper's impact.